# LOCAL VS. GLOBAL INTERPRETABILITY: A COMPUTATIONAL PERSPECTIVE

## ABSTRACT

The local and global interpretability of various ML models has been studied extensively in recent years. Yet despite significant progress in the field, many of the known results are either informal or lack sufficient mathematical rigor. In this work, we propose a framework based on computational complexity theory to systematically evaluate the local and global interpretability of different ML models. In essence, our framework examines various forms of explanations that can be computed either locally or globally and assesses the computational complexity involved in generating them. We begin by rigorously studying global explanations, and establish: (i) a duality relationship between local and global forms of explanations; and (ii) the inherent uniqueness associated with certain global forms of explanations. We then proceed to evaluate the computational complexity associated with these forms of explanations, with a particular emphasis on three model types usually positioned at the extremes of the interpretability spectrum: (i) linear models; (ii) decision trees; and (iii) neural networks. Our findings reveal that, assuming standard complexity assumptions such as $P \neq NP$, computing global explanations is computationally more difficult for linear models than for their local counterparts. Surprisingly, this phenomenon is not universally applicable to decision trees and neural networks: in certain scenarios, computing a global explanation is actually more tractable than computing a local one. We consider these results as compelling evidence of the importance of analyzing ML explainability from a computational complexity perspective, as the means of gaining a deeper understanding of the inherent interpretability of diverse ML models.

## 1 INTRODUCTION

*Interpretability* is becoming an increasingly important aspect of ML models, as it plays a key role in ensuring the safety, transparency and fairness of these models (Doshi-Velez & Kim (2017)). The ML community has been studying two notions of interpretability: *global interpretability*, aimed at understanding the overall decision logic of an ML model; and *local interpretability*, aimed at understanding specific decisions made by that model (Zhang et al. (2021); Du et al. (2019)). Various claims have been made concerning the relative interpretability of various ML models (Gilpin et al. (2018)). For instance, it has been proposed that linear classifiers inherently possess local interpretability, but lack global interpretability; whereas decision trees are acknowledged to possess both local and global interpretability (Molnar (2020)). Although such insights are intuitively helpful, they often lack mathematical rigor.

Here, we argue that a more formal basis is required in order to establish a sound theoretical foundation for assessing interpretability. To do so, we propose to study interpretability through the lens of *computational complexity* theory. We seek to study different notions of model explanations, and determine the computational complexity of obtaining them — in order to measure the interpretability of different models. Intuitively, the lower the complexity, the more interpretable the model is with respect to that form of explanation. Recent work provided new insights in this direction (Barceló et al. (2020); Marques-Silva et al. (2020); Arenas et al. (2022)), but focused mainly on *local* forms of explanations — thus contributing to formally measuring the *local* interpretability across various contexts, rather than addressing the issue of global interpretability.

**Our contributions.** We present a formal framework for evaluating the local and global interpretability of ML models. The framework rigorously assesses the computational complexity required to obtain various explanation forms, either local (pertaining to a specific instance **x**) or global (pertaining to any possible **x**). Consequently, it affords insights into the local and global interpretability level of the various models and explanation forms. We focus on the following forms of explanations:

1. **Sufficient reasons:** subsets of input features that are sufficient for determining the model's output. Global sufficient reasons imply that this subset always determines the result, whereas local sufficient reasons imply that this subset determines the classification under a partial assignment of some input. We also study the problem of obtaining sufficient reasons *of smallest cardinality*.

2. **Redundant features:** input features that do not contribute to a specific prediction, or, in the global case, do not contribute to any prediction.

3. **Necessary features:** input features *essential* for determining a specific prediction, or, in the global case, for determining any prediction.

4. **Completion count:** the *relative portion* of assignments that maintain a prediction, given that we fix some subset of features. This form relates to the *probability* of obtaining a prediction, and can be computed either locally or globally.

We present several novel insights concerning the overarching characteristics of these types of explanations, culminating in: (i) a duality relationship between local and global forms of explanations; and (ii) a result on the *uniqueness* of certain global forms of explanations, in stark contrast to the exponential abundance of their local counterparts.

In addition, we study the computational complexity of obtaining these forms of explanations, for various ML models. We focus on models that are frequently mentioned in the literature as being at the two ends of the interpretability spectrum: (i) decision trees; (ii) linear models; and (iii) neural networks. Our results allow us to determine whether models are more *locally interpretable* (it is easier to interpret them locally than globally), or more *globally interpretable* (the opposite case); and also establish a complexity hierarchy of explaining various models.

In some cases, our results rigorously justify prior claims. For example, we establish that linear models are indeed easier to interpret locally than globally (Molnar (2020)): while obtaining local sufficient reasons can be performed in polynomial time, obtaining *global* sufficient reasons for these models is coNP-Complete. In other cases, our results actually defy intuition. For example, we discover that neural networks and decision trees are easier to interpret globally than locally, for some explanation forms: e.g., minimally sized *global* sufficient reasons can be obtained in polynomial time for decision trees, but obtaining their *local* counterparts is NP-Complete. A similar pattern is found for neural networks, and again both for neural networks and decision trees when concerning the evaluation of redundant features. These results highlight the benefits of a rigorous study of these topics.

Due to space constraints, we include a brief outline of the proofs of our various claims within the paper, and relegate the complete proofs to the appendix.

## 2 PRELIMINARIES

**Complexity Classes.** The paper assumes basic familiarity with the common complexity classes of polynomial time (PTIME) and nondeterministic polynomial time (NP, co-NP). We also mention classes of the second order of polynomial hierarchy, i.e., $\Sigma_2^P$, which describes the set of problems that could be solved in NP given an oracle that solves problems of co-NP in constant time, and $\Pi_2^P$, which describes the set of problems that could be solved in co-NP given an oracle that solves problems of NP in constant time. Both NP and co-NP are contained in both $\Sigma_2^P$ and $\Pi_2^P$, and it is also widely believed that this containment is strict i.e., PTIME$\subsetneq$ NP, co-NP$\subsetneq \Sigma_2^P$, $\Pi_2^P$ (Arora & Barak (2009)) which is a consequence of the assumption that PTIME$\neq$NP. We also discuss the class #P, which corresponds to the total number of accepting paths of a polynomial-time nondeterministic Turing machine. It is also widely believed that #P strictly contains the second order of the polynomial hierarchy, i.e., that $\Sigma_2^P, \Pi_2^P \subsetneq$ #P (Arora & Barak (2009)).

**Domain.** We assume a set of $n$ features $\mathbf{x} := (x_1, \ldots, x_n)$, and use $\mathbb{F} := \{0, 1\}^n$ to denote the entire feature space. Our goal is to interpret the prediction of a classifier $f : \mathbb{F} \to \{0, 1\}$. In the *local* case, we seek the reason behind the prediction $f(\mathbf{x})$ made for a specific instance $\mathbf{x}$. In the *global* case, we seek to explain the general behavior of $f$. (We follow common practice and focus on Boolean input and output values, to simplify the presentation (Arenas et al. (2021); Wäldchen et al. (2021); Barceló et al. (2020)). We note, however, that many of our results carry over to the real-valued case, as well.)

**Explainability Queries.** To address interpretability's abstract nature, prior work often uses a construct called an *explainability query* (Barceló et al. (2020)), denoted $Q$, which defines an explanation of a specific type. As prior work focused mainly on *local* explanation forms, the input of $Q$ is usually comprised of both $f$ and $\mathbf{x}$, and its output is an answer providing information regarding the interpretation of $f(\mathbf{x})$. For any given explainability query $Q$, we define its corresponding *global form of explanation* as $G$-$Q$. In contrast to $Q$, the input of $G$-$Q$ does not include any specific instance $\mathbf{x}$ and the output conditions hold for *any* $\mathbf{x} \in \mathbb{F}$ rather than only for $\mathbf{x}$. We provide the full formalization of each local and global explainability query in Section 3.

# 3 LOCAL AND GLOBAL EXPLANATION FORMS

Although model interpretability is subjective, there are several commonly used notions of local and global explanations, on which we focus here:

**Sufficient reasons.** A *local* sufficient reason is a subset of features, $S \subseteq \{1, \ldots, n\}$. When the features in $S$ are fixed to their corresponding values in $\mathbf{x}$, the prediction is determined to be $f(\mathbf{x})$, regardless of other features' assignments. Formally, S is a sufficient reason with respect to $\langle f, \mathbf{x} \rangle$ if for any $\mathbf{x} \in \mathbb{F}$ it holds that $f(\mathbf{x}_S; \mathbf{z}_{\bar{S}}) = f(\mathbf{x})$. Here, $(\mathbf{x}_S; \mathbf{z}_{\bar{S}})$ represents an assignment where the values of elements of $S$ are taken from $\mathbf{x}$, and the remaining values $\overline{S}$ are taken from $\mathbf{z}$.

A set $S \subseteq \{1, \ldots, n\}$ is a *global* sufficient reason of $f$ if it is a local sufficient reason for all $\mathbf{x}$. More formally: for any $\mathbf{x}, \mathbf{z} \in \mathbb{F}$, it holds that $f(\mathbf{x}_S; \mathbf{z}_{\bar{S}}) = f(\mathbf{x})$. We denote $\text{suff}(f, S, \mathbf{x}) = 1$ when $S$ is a local sufficient reason of $\langle f, \mathbf{x} \rangle$, and $\text{suff}(f, S, \mathbf{x}) = 0$ otherwise. Similarly, we denote $\text{suff}(f, S) = 1$ when $S$ is a *global* sufficient reason of $f$, and $\text{suff}(f, S) = 0$ otherwise. This leads us to our first explainability query:

---
**CSR (Check Sufficient Reason)**:
**Input**: Model $f$, input $\mathbf{x}$, and subset of features $S$
**Output**: *Yes* if $\text{suff}(S, f, \mathbf{x}) = 1$, and *No* otherwise

---

To differentiate between the local and global setting, we use *G-CSR* to refer to the explainability query that checks whether $S$ is a *global* sufficient reason of $f$. Due to space limitations, we relegate the full formalization of global queries to the appendix.

A common notion in the literature suggests that smaller sufficient reasons (i.e., with smaller $|S|$) are more meaningful than larger ones (Ribeiro et al. (2018); Halpern & Pearl (2005)). Consequently, it is interesting to consider *cardinally minimal sufficient reasons* (also known as *minimum sufficient reasons*), which are computationally harder to obtain (Barceló et al. (2020)).

---
**MSR (Minimum Sufficient Reason)**:
**Input**: Model $f$, input $\mathbf{x}$, and integer k
**Output**: *Yes* if there exists some $S$ such that $\text{suff}(S, f, \mathbf{x}) = 1$ and $|S| \leq k$, and *No* otherwise

---

Similarly, *G-MSR* denotes the explainability query for obtaining global cardinally minimal sufficient reasons. This notion aligns with the optimization goal of many *feature selection* tasks (Wang et al. (2015)), where one seeks to select minimal subsets of features that globally determine a prediction.

**Necessity and redundancy.** When interpreting a model, it is common to measure the importance of each feature to a prediction. Here we analyze the complexity of identifying the two extreme cases: features that are either *necessary* or *redundant* to a prediction (Huang et al. (2023)). These kinds of features are useful in the study of some notions of *fairness* (Ignatiev et al. (2020a)): necessary features can be seen as biased features, whereas redundant features are protected features that should

not be used for decision-making, such as gender, age, etc. (Arenas et al. (2021); Darwiche & Hirth (2020)).

Formally, we say that a feature $i$ is necessary with respect to $\langle f, \mathbf{x} \rangle$ if it is contained in *all* sufficient reasons of $\langle f, \mathbf{x} \rangle$. This implies that removing $i$ from any sufficient reason $S$ causes it to cease being sufficient; i.e., for any $S \subseteq \{1, \ldots, n\}$ it holds that $\mathrm{suff}(f, \mathbf{x}, S) = 1 \rightarrow \mathrm{suff}(f, \mathbf{x}, S \setminus \{i\}) = 0$.

In the global case, we would seek to determine whether feature $i$ is *globally necessary* to $f$, meaning it is necessary to *all* instances of $\langle f, \mathbf{x} \rangle$. Formally, for any $\mathbf{x} \in \mathbb{F}$ and for any $S \subseteq \{1, \ldots, n\}$ it holds that $\mathrm{suff}(f, \mathbf{x}, S) = 1 \rightarrow \mathrm{suff}(f, \mathbf{x}, S \setminus \{i\}) = 0$.

---

**FN (Feature Necessity)**:
**Input**: Model $f$, input $\mathbf{x}$, and integer $i$
**Output**: *Yes* if $i$ is necessary with respect to $\langle f, \mathbf{x} \rangle$, and *No* otherwise

---

The *G-FN* formalization (along with other global queries in this section) appears in the appendix.

Conversely, a feature $i$ is said to be *locally redundant* with respect to $\langle f, \mathbf{x} \rangle$ if removing it from any sufficient reason $S$ does not affect $S$'s sufficiency. Formally, for any $S \subseteq \{1, \ldots, n\}$ it holds that $\mathrm{suff}(f, \mathbf{x}, S) = 1 \rightarrow \mathrm{suff}(f, \mathbf{x}, S \setminus \{i\}) = 1$:

---

**FR (Feature Redundancy)**:
**Input**: Model $f$, input $\mathbf{x}$, and integer $i$.
**Output**: *Yes*, if $i$ is redundant with respect to $\langle f, \mathbf{x} \rangle$, and *No* otherwise.

---

We say that a feature is *globally redundant* if it is locally redundant with respect to all inputs; i.e., for any $\mathbf{x} \in \mathbb{F}$ and $S \subseteq \{1, \ldots, n\}$ it holds that $\mathrm{suff}(f, \mathbf{x}, S) = 1 \rightarrow \mathrm{suff}(f, \mathbf{x}, S \setminus \{i\}) = 1$.

**Counting completions.** One final common form of explainability is based on exploring the relative *portion* of assignment completions that maintain a specific classification, which relates to the *probability* that a prediction remains the same, assuming the other features are uniformly and independently distributed. We define the local completion count $c$ of $S$ as the relative portion of completions which maintain the prediction of $f(\mathbf{x})$:

$$c(S, f, \mathbf{x}) := \frac{|\{\mathbf{z} \in \{0,1\}^{|\overline{S}|}, f(\mathbf{x}_S; \mathbf{z}_{\bar{S}}) = f(\mathbf{x})\}|}{|\{\mathbf{z} \in \{0,1\}^{|\overline{S}|}|} \tag{1}$$

In the global completion count case, we count the number of completions for all possible assignments $\mathbf{x} \in \mathbb{F}$:

$$c(S, f) := \frac{|\{\mathbf{x} \in \mathbb{F}, \mathbf{z} \in \{0,1\}^{|\overline{S}|}, f(\mathbf{x}_S; \mathbf{z}_{\bar{S}}) = f(\mathbf{x})\}|}{|\{\mathbf{x} \in \mathbb{F}, \mathbf{z} \in \{0,1\}^{|\overline{S}|}|} \tag{2}$$

---

**CC (Count Completions)**:
**Input**: Model f, input x, and subset of features S
**Output**: The completion count $c(S, f, \mathbf{x})$

---

We acknowledge that other explanation forms can be used, and do not argue that one form is superior to others; rather, our goal is to study some local and global versions of common explanation forms as a means of assessing the local and global interpretability of different ML models.

## 4 PROPERTIES OF GLOBAL EXPLANATIONS

We now present several novel results concerning the characteristics of global explanations, and in Section 5 we subsequently illustrate how these results significantly affect the complexity of actually computing such explanations.

### 4.1 DUALITY OF LOCAL AND GLOBAL EXPLANATIONS

Our analysis shows that there exists a dual relationship between local and global explanations. To better understand this relationship, we make use of the definition of *contrastive reasons*, which

describes subsets of features that, when altered, may cause the classification to change. Formally, a subset of features $S$ is a contrastive reason with respect to $\langle f, \mathbf{x} \rangle$ iff there exists some $\mathbf{z} \in \mathbb{F}$ such that $f(\mathbf{x}_{\bar{S}}; \mathbf{z}_S) \neq f(\mathbf{x})$.

While sufficient reasons provide answers to "*why?*" questions, i.e., "why was $f(\mathbf{x})$ classified to class $i$?", contrastive reasons seek to provide answers to "*why not?*" questions. Clearly, $S$ is a sufficient reason of $\langle f, \mathbf{x} \rangle$ iff $\overline{S}$ is *not* a contrastive reason of $\langle f, \mathbf{x} \rangle$. Contrastive reasons are also well related to necessity. This is shown by the following theorem, whose proof appears in the appendix:

**Theorem 1** *A feature $i$ is necessary with respect to $\langle f, \boldsymbol{x} \rangle$ if and only if $\{i\}$ is a contrastive reason of $\langle f, \boldsymbol{x} \rangle$.*

We can similarly define a *global contrastive reason* as a subset of features that may cause a misclassification for any possible input. Formally, for any $\mathbf{x} \in \mathbb{F}$ there exists some $\mathbf{z} \in \mathbb{F}$ such that $f(\mathbf{x}_{\bar{S}}; \mathbf{z}_S) \neq f(\mathbf{x})$. This leads to a first dual relationship between local and global explanations:

**Theorem 2** *Any global sufficient reason of $f$ intersects with all local contrastive reasons of $\langle f, \boldsymbol{x} \rangle$, and any global contrastive reason of $f$ intersects with all local sufficient reasons of $\langle f, \boldsymbol{x} \rangle$.*

This formulation can alternatively be expressed through the concept of *hitting sets* (additional details appear in the appendix). In this context, global sufficient reasons correspond to hitting sets of local contrastive reasons, while local contrastive reasons correspond to hitting sets for global sufficient reasons. It follows that the minimum hitting set (MHS; see appendix) aligns with cardinally minimal reasons. Formally:

**Theorem 3** *The MHS of all local contrastive reasons of $\langle f, \boldsymbol{x} \rangle$ is a cardinally minimal global sufficient reason of $f$, and the MHS of all local sufficient reasons of $\langle f, \boldsymbol{x} \rangle$ is a cardinally minimal global contrastive reason of $f$.*

## 4.2 Uniquness of Global Explanations

As stated earlier, small sufficient reasons are often assumed to provide a better interpretation than larger ones. Consequently, we are interested in *minimal* sufficient reasons, i.e., explanation sets that cease to be sufficient reasons as soon as even one feature is removed from them. We note that *minimal* sufficient reasons are not necessarily cardinally minimal, and we can also consider *subset minimal* sufficient reasons (alternatively referred to as locally minimal). The choice of the terms *cardinally minimal* and *subset minimal* is deliberate, to reduce confusion with the concepts of global and local explanations.

A greedy approach for computing subset minimal explanations appears in Algorithm 1. It starts with the entire set of features and gradually attempts to remove them until converging to a *subset minimal* sufficient reason. Notably, the validation step at Line 3 within the algorithm, which determines the sufficiency of a feature subset, is not straightforward. In Section 5, we delve into a detailed discussion of the computational complexities associated with this process.

---

**Algorithm 1** Local Subset Minimal Sufficient Reason

**Input** $f, \mathbf{x}$

1: $S \leftarrow \{1, \ldots, n\}$
2: **for each** $i \in \{1, ..., n\}$ by some arbitrary ordering **do**
3:     **if** $\text{suff}(f, S \setminus \{i\}, \mathbf{x}) = 1$ **then**
4:         $S \leftarrow S \setminus \{i\}$
5:     **end if**
6: **end for**
7: **return** $S$                    $\triangleright$ $S$ is a subset minimal *local* sufficient reason

---

While Algorithm 1 converges to a subset-minimal local sufficient reason, it is not necessarily a cardinally minimal sufficient reason. This is due to the algorithm's strong sensitivity to the order in which we iterate over features (Line 2). The number of subset-minimal and cardinally minimal sufficient reasons depends on the function $f$. Nevertheless, it can be shown that their prevalence is, in the worst-case, *exponential* in the number of features $n$:

**Proposition 1** *There exists some $f$ and some $\boldsymbol{x} \in \mathbb{F}$ such that there are $2^{\lfloor \frac{n}{2} \rfloor}$ local subset minimal or cardinally minimal sufficient reasons of $\langle f, \boldsymbol{x} \rangle$.*

A similar, greedy approach for *global* explanations appears in Algorithm 2:

---
**Algorithm 2** Global Subset Minimal Sufficient Reason
---
**Input** $f$
 1: $S \leftarrow \{1, \ldots, n\}$
 2: **for each** $i \in \{1, ..., n\}$ by some arbitrary ordering **do**
 3:      **if** $\text{suff}(f, S \setminus \{i\}) = 1$ **then**
 4:          $S \leftarrow S \setminus \{i\}$
 5:      **end if**
 6: **end for**
 7: **return** $S$              ▷ $S$ is a subset minimal *global* sufficient reason

---

Given that the criteria for a subset of features to constitute a *global* sufficient reason are more stringent than those for the local case, it is natural to ask whether they are also exponentially abundant. To start addressing this question, we establish the following proposition:

**Proposition 2** *If $S_1$ and $S_2$ are two global sufficient reasons of $f$, then $S_1 \cap S_2 = S \neq \emptyset$, and $S$ is a global sufficient reason of $f$.*

*Proof Sketch.* Using the duality property of Theorem 2, it is possible to conclude that $S_1 \cap S_2 \neq \emptyset$. For the second part of the claim, whenever $S_1 \subseteq S_2$ or $S_2 \subseteq S_1$, the proof is trivial. When that is not the case, we observe that $S = S_1 \cap S_2$ must be a local sufficient reason with respect to $\langle f, \mathbf{x} \rangle$ for any $\mathbf{x} \in \mathbb{F}$, and is hence a global sufficient reason with respect to $f$.

From Proposition 2 now stems the following theorem:

**Theorem 4** *There exists one unique subset-minimal global sufficient reason of $f$.*

Thus, while the local form of explanation presents us with a worst-case scenario of an *exponential* number of minimal explanations, the global form, on the other hand, offers only a single, unique minimal explanation. As we demonstrate later, this distinction causes significant differences in the complexity of computing such explanations. We can now derive the following corollary:

**Proposition 3** *For any possible ordering of features in Line 4.2 of Algorithm 2, Algorithm 2 converges to the same global sufficient reason.*

The uniqueness of global subset-minimal sufficient reasons also carries implications for the assessment of feature necessity and redundancy, as follows:

**Proposition 4** *Let $S$ be the subset minimal global sufficient reason of $f$. For all $i$, $i \in S$ if and only if $i$ is locally necessary for some $\langle f, \boldsymbol{x} \rangle$, and $i \in \overline{S}$ if and only if $i$ is globally redundant for $f$.*

In other words, subset $S$, which is the unique minimal global sufficient reason of $f$, categorizes the features into two possible sets: those *necessary* to a specific instance $\mathbf{x}$, and those that are *globally redundant*. This fact is further exemplified by the subsequent corollary:

**Proposition 5** *Any feature $i$ is either locally necessary for some $\langle f, \boldsymbol{x} \rangle$, or globally redundant for $f$.*

## 5   THE COMPUTATIONAL COMPLEXITY OF GLOBAL INTERPRETATION

We seek to conduct a comprehensive analysis of the computational complexity of computing local and global explanations, of the forms discussed in Section 3. We perform this analysis on three *classes* of models: free binary decision diagrams (FBDDs), which are a generalization of decision trees, Perceptrons, and multi-layer Perceptrons (MLPs) with reLU activation units. A full formalization of these model classes is provided in the appendix.

We use $Q(C)$ (respectively, $G$-$Q(C)$) to denote the computational problem of solving the *local* (respectively, *global*) explainability query $Q$ on models of class $C$. Table 1 summarizes our results, and indicates the complexity classes for model class and explanation type pairs.

Table 1: Complexity classes for pairs of explainability queries and model classes. Cells highlighted in blue are the result of novel proofs, presented here; while the rest are based on prior work.

| | **FBDDs** | | **MLPs** | | **Perceptrons** | |
|---|---|---|---|---|---|---|
| | Local | Global | Local | Global | Local | Global |
| **CSR** | PTIME | PTIME | coNP-C | coNP-C | PTIME | coNP-C |
| **MSR** | NP-C | PTIME | $\Sigma_2^P$-C | coNP-C | PTIME | coNP-C |
| **CC** | PTIME | PTIME | #P-C | #P-C | #P-C | #P-C |
| **FR** | coNP-C | PTIME | $\Pi_2^P$-C | coNP-C | coNP-C | coNP-C |
| **FN** | PTIME | PTIME | PTIME | coNP-C | PTIME | PTIME |

As these results demonstrate, there is often a *strict disparity* in computational effort between calculating local and global explanations, emphasizing the need for distinct assessments of local and global forms. We further study this disparity and investigate the *comparative* computational efforts required for local and global explanations across various models and forms of explanations. This examination enables us to address the fundamental question of whether certain models exhibit a higher degree of interpretability at a global level compared to their interpretability at a local level, within different contextual scenarios. In Section 5.1, we introduce a framework to investigate this question. In section 5.2, we delve into the technical aspects of the required reductions for proving these complexity classes.

## 5.1 LOCAL VS. GLOBAL INTERPRETABILITY

We say that a model is more *locally interpretable* for a given explanation type if computing the local form of that explanation is strictly easier than computing the global form, and say that it is more *globally interpretable* in the opposite case. Formally put:

**Definition 1** *Let $Q$ denote an explainability query and $C$ a class of models, and suppose $Q(C)$ is in class $K_1$ and $G$-$Q(C)$ is in class $K_2$. Then:*

1. *$C$ is strictly more locally c-interpretable with respect to $Q$ iff $K_1 \subsetneq K_2$ and $G$-$Q(C)$ is hard for $K_2$.*

2. *$C$ is strictly more globally c-interpretable with respect to $Q$ iff $K_2 \subsetneq K_1$ and $Q(C)$ is hard for $K_1$.*

We begin by studying the Perceptron model. As depicted in Table 1, there exists a disparity between the local and global forms for the *CSR* and *MSR* queries: while the local forms can be obtained in polynomial time, obtaining the global forms is coNP-Complete. This leads us to our first corollary:

**Theorem 5** *Perceptrons are strictly more locally c-interpretable with respect to CSR and MSR.*

The fact that a class of models is more locally interpretable may seem intuitive. Nevertheless, it is rather surprising that, in certain instances, acquiring global explanations can be comparatively simpler. Notably, our findings demonstrate that this surprising result holds for both FBDDs and MLPs. For FBDDs, the minimum-sufficient-reason (*MSR*) and feature redundancy (*FR*) queries *are easier to obtain at the global level*, leading us to our second corollary:

**Theorem 6** *FBDDs are strictly more globally c-interpretable with respect to MSR and FR.*

As we later discuss in Section 5.2, this result stems from the fact that global cardinally minimal sufficient reasons are *unique*, making them easier to obtain for some models. Additionally, due to the

relationship between global cardinally minimal sufficient reasons and globally redundant features (Theorem 4), the uniqueness property can also affect the complexity of *G-FR* queries. Notice that this is the case for FBDDs but was *not* the case for Perceptrons, since the complexity for checking global sufficient reasons in Perceptrons (*G-CSR*) was higher to begin with, consequently affecting the complexities of the *G-MSR* and *G-FR* queries.

Finally, in the case of MLPs, our analysis does not provide a clear-cut division between local and global interpretability.

**Theorem 7** *MLPs are (i) strictly more globally c-interpretable with respect to MSR and FR, and (ii) strictly more locally c-interpretable with respect to FN.*

## 5.2 Computational Classes for Global Explainability Queries

The outcomes outlined in the preceding section stem directly from our novel proofs of the associated *global* forms of explainability queries. In this section, we delve further into the associated complexity classes and discuss the reductions used to obtain them.

**Complexity of Global Sufficient Reasons.** We start off with providing proof sketches of the complexity classes associated with *checking* if a subset of features is a sufficient reason:

**Proposition 6** *G-CSR is (i) coNP-Complete for MLPs/Perceptrons, and (ii) in PTIME for FBDDs.*

*Proof Sketch.* Membership in coNP holds since one can guess certificates $\mathbf{x} \in \mathbb{F}$ and $\mathbf{z} \in \mathbb{F}$ and validate whether $S$ is not sufficient. For Perceptrons, we propose a reduction from the $\overline{SSP}$ (subset-sum problem), which is coNP-Complete. Hardness for Perceptrons clearly also holds for MLPs, but we nevertheless show that it can be obtained via a reduction from (local) *CSR* for MLPs, which is coNP-Complete. For FBDDs, we provide a polynomial algorithm.

We then move on to assess the complexity associated with obtaining *cardinally minimal* global sufficient reasons. In contrast to the local scenario where obtaining cardinally minimal sufficient reasons is harder (Barceló et al. (2020)), this difference does not persist in the global context:

**Proposition 7** *G-MSR is (i) coNP-Complete for MLPs/Perceptrons and (ii) in PTIME for FBDDs.*

*Proof Sketch.* Membership is a consequence of Proposition 4, which shows that any feature that is contained in the subset minimal global sufficient reason is necessary for some $\langle f, \mathbf{x} \rangle$, or is globally redundant otherwise. We hence can guess $n$ assignments $\mathbf{x}^1, \ldots, \mathbf{x}^n$, and for each feature $i \in \{1, \ldots n\}$, validate whether $i$ is locally necessary for $\langle f, \mathbf{x}^i \rangle$, and whether this holds for more than $k$ features. We prove hardness for MLPs/Perceptrons using a similar reduction to *CSR*. Given that *CSR* for FBDDs can be solved in PTIME, it follows from Algorithm 2 that *MSR* is in PTIME.

**Complexity of Global Necessity and Redundancy.** We provide proof sketches for the complexity of validating whether input features are globally redundant or necessary, starting with redundancy:

**Proposition 8** *G-FR is (i) coNP-Complete for MLPs/Perceptrons, and (ii) in PTIME for FBDDs.*

*Proof Sketch.* From Theorem 1 and Proposition 5, we can conclude that $i$ is *not* globally redundant iff $\{i\}$ is contrastive for some $\langle f, \mathbf{x} \rangle$. This property is useful for demonstrating membership and hardness for MLPs/Perceptrons and for devising a polynomial algorithm for FBDDs.

For global necessity, we derive different complexity classes:

**Proposition 9** *G-FN is (i) coNP-Complete for MLPs, and (ii) in PTIME for Perceptrons and FBDDs.*

*Proof Sketch.* Membership in coNP can be obtained using Theorem 1. We prove hardness for MLPs by reducing from *TAUT*, a classic coNP-Complete problem that checks whether a Boolean formula is a tautology. For Perceptrons and FBDDs we suggest polynomial algorithms whose correctness is derived from Theorem 1.

These results imply an intriguing consequence regarding the complexity of validating necessity and redundancy in the specific case of MLPs:

**Observation 1** *For MLPs, global necessity (G-FN) is strictly harder than local necessity (FN), whereas global redundancy (G-FR) is strictly less hard than local redundancy (FR).*

Another interesting insight from the previous theorems is the comparison between MLPs and Perceptrons. Since Perceptrons are a specific case of MLPs with only one layer, analyzing the computational complexity difference between them can provide insights into the influence of hidden layers on model intricacy. Our findings indicate that for some queries while hidden layers affect *local* queries, they do not impact *global* queries.

**Observation 2** *Obtaining CSR, MSR, and FR is strictly harder for MLPs compared to MLPs with no hidden layers. However, this disparity does not exist for G-CSR, G-MSR, and G-FR.*

**Complexity of Global Count Completions.** Finally, We offer proof sketches for the global *CC* queries. Unlike the previous queries, here the complexity classes for global configurations remain akin to their local counterparts.

**Proposition 10** *G-CC is (i) $\#P$-Complete for MLPs and Perceptrons and (ii) in PTIME for FBDDs.*

*Proof Sketch.* Membership in $\#P$ is straightfowrad. For the hardness in case of Perceptrons/MLPs, we reduce from (local) *CC* of Perceptrons which is $\#P$-complete. For FBDDs, we offer a polynomial algorithm.

Our work also introduces new complexity classes for local explanation configurations, as shown in Table 1. For brevity, we include the proofs in the appendix (Proposition 11).

## 6 RELATED WORK

Our work contributes to a line of research on formal explainable AI (XAI) (Marques-Silva et al. (2020)), which focuses on obtaining explanations with mathematical guarantees. Several papers have already explored the computational complexity of obtaining such explanations (Barceló et al. (2020); Wäldchen et al. (2021); Arenas et al. (2022)); however, these efforts focused on local explanations, whereas we focus on both local and global explanations.

Some of the terms leveraged in our work were discussed in the literature: "sufficient reasons" are also known as *abductive explanations* (Ignatiev et al. (2019)), while minimal sufficient reasons are sometimes referred as *prime implicants* in Boolean formulas (Darwiche & Marquis (2002)). A notion similar to the *CC* query is the $\delta$-relevant set (Wäldchen et al. (2021); Izza et al. (2021)), which asks whether the completion count exceeds a threshold $\delta$. Similar duality properties to the ones studied here were shown to hold considering the relationship between local sufficient and contrastive reasons (Ignatiev et al. (2020b)), and the relationship between absolute sufficient reasons and adversarial attacks (Ignatiev et al. (2019)). Minimal *absolute* sufficient reasons refer to subsets that are the smallest among all possible inputs and rely on partial input assignments. In our global sufficient reason definition, we do not rely on particular inputs or partial assignments.

The necessity and redundancy queries that we discussed were studied previously (Huang et al. (2023)) and are related to forms of fairness (Ignatiev et al. (2020a)). In this context, necessity is related to biased features, and redundant features are related to protected features (Arenas et al. (2021); Darwiche & Hirth (2020)). Of course, there exist many notions of bias and fairness (Mehrabi et al. (2021)).

## 7 CONCLUSION

We present a theoretical framework using computational complexity theory to assess both local and global interpretability of ML models. Our work uncovers new insights, including a duality relationship between local and global explanations and the uniqueness inherent in some global explanation forms. We also offer novel proofs for complexity classes related to global explanations and demonstrate how our insights impact these classes. This allows us to *formally measure interpretability* for different models across various contexts. We apply these insights to commonly evaluated ML models, including linear models, decision trees, and neural networks.

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

# Appendix

The appendix contains formalizations and proofs that were mentioned throughout the paper:

## A    GLOBAL FORMS OF MODEL EXPLANATIONS

In this section, we present the *global* forms of the explainability queries previously mentioned, which were initially formulated in the paper for their local configuration.

---

**G-CSR (*Global* Check Sufficient Reason)**:
**Input**: Model $f$, and subset of features $S$.
**Output**: *Yes*, if $S$ is a global sufficient reason of $f$ (i.e., $\text{suff}(f, S) = 1$), and *No* otherwise.

---

**G-MSR (*Global* Minimum Sufficient Reason)**:
**Input**: Model $f$, and integer k.
**Output**: *Yes*, if there exists a global sufficient reason $S$ for $f$ (i.e., $\text{suff}(f, S) = 1$) such that $|S| \leq k$, and *No* otherwise.

---

**G-FR (*Global* Feature Redundancy)**:
**Input**: Model $f$, and integer $i$.
**Output**: *Yes*, if $i$ is globally redundant with respect to $f$, and *No* otherwise.

---

**G-FN (*Global* Feature Necessity)**:
**Input**: Model $f$, and integer $i$.
**Output**: *Yes*, if $i$ is globally necessary with respect to $f$, and *No* otherwise.

---

**G-CC (*Global* Count Completions)**:
**Input**: Model $f$, and subset $S$.
**Output**: The global completion count $c(S, f)$

---

## B    MODEL CLASSES

In the next appendix, we describe in detail the various model classes that were taken into account within this work:

**Free Binary Decision Diagram (FBDD).** A *BDD* is a graph-based model that represents a Boolean function $f : \mathbb{F} \to \{0, 1\}$ (Lee (1959)). The arbitrary Boolean function is realized by an acyclic (directed) graph, for which the following holds: (i) every internal node $v$ corresponds with a single binary input feature $(1, \ldots, n)$; (ii) every internal node $v$ has exactly two output edges, that represent the values $\{0, 1\}$ assigned to $v$; (iii) each leaf node corresponds to either a *True*, or *False*, label; and (iv) every variable appears *at most* once, along every path $\alpha$ of the BDD.

Hence, any assignment to the inputs $\mathbf{x} \in \mathbb{F}$ corresponds to one unique path $\alpha$ from the BDD's root to one of its leaf nodes. We denote $f(\mathbf{x}) := 1$ if the label of the leaf node is true, and $f(\mathbf{x}) := 0$ if it is false. Moreover, we regard the size of a BDD (i.e., $|f|$ ) to be the overall number of edges in the BDD's graph. In this work, we focus on the popular variant of *Free BDDs* (FBDDs) models, for which different paths, $\alpha, \alpha'$ are allowed to have different orderings of the input variables $\{1, \ldots, n\}$ and on every path $\alpha$ no two nodes have the same label. A *decision tree* can be essentially described as an FBDD whose foundational graph structure is a tree.

**Multi-Layer Perceptron (MLP).** A Multi-Layer Perceptron (Gardner & Dorling (1998); Ramchoun et al. (2016)) $f$ with $t - 1$ hidden layers ($g^j$ for $j \in \{1, \ldots, t - 1\}$) and a single output layer ($g^t$), is recursively defined as follows: $g^{(j)} := \sigma^{(j)}(g^{(j-1)}W^{(j)} + b^{(j)})$   ($j \in \{1, \ldots, t\}$), given $t$ weight matrices $W^{(1)}, \ldots, W^{(t)}$, $t$ bias vectors $b^{(1)}, \ldots, b^{(t)}$, and also $t$ activation functions $f^{(1)}, \ldots, f^{(t)}$.

The MLP $f$ outputs the value $f := g^{(t)}$, while $g^{(0)} := \mathbf{x} \in \{0,1\}^n$ is the input layer that receives the input of the model. The biases and weight matrices are defined by a series of positive values $d_0, \ldots, d_t$ that represent their dimensions. Furthermore, we assume that all the weights and biases possess rational values, denoted as $W^{(j)} \in \mathbb{Q}^{d_{j-1} \times d_j}$ and $b^{(j)} \in \mathbb{Q}^{d_j}$, which have been acquired during the training phase. Due to our focus on *binary* classifiers over $\{1, \ldots, n\}$, it necessarily holds that: $d_0 = n$ and $d_t = 1$. In this work, we focus on the popular $reLU(x) = \max(0, x)$ activation function, with the exception of the single activation in the last layer, that is typically a sigmoid function. Nonetheless, given our emphasis on post-hoc interpretability, it is without loss of generality that we may assume the last activation function is represented by the step function, i.e., $step(\mathbf{z}) = 1 \iff \mathbf{z} \geq 0$.

**Perceptron.** A Perceptron (Ralston et al. (2003)) is a single-layered MLP (i.e., $t = 1$): $f(\mathbf{x}) = step((\mathbf{w} \cdot \mathbf{x}) + b)$, for $b \in \mathbb{Q}$ and $\mathbf{w} \in \mathbb{Q}^{n \times d_1}$. Thus, for a Perceptron $f$ the following holds w.l.o.g.: $f(\mathbf{x}) = 1 \iff (\mathbf{w} \cdot \mathbf{x}) + b \geq 0$.

## C  THE DUALITY OF LOCAL AND GLOBAL EXPLANATIONS

**Minimum Hitting Set (MHS).** Given a collection $\mathbb{S}$ of sets from a universe U, a hitting set $h$ for $\mathbb{S}$ is a set such that $\forall S \in \mathbb{S}, h \cap S \neq \emptyset$. A hitting set $h$ is said to be *minimal* if none of its subsets is a hitting set, and *minimum* when it has the smallest possible cardinality among all hitting sets.

**Theorem 1** *A feature $i$ is necessary w.r.t $\langle f, \boldsymbol{x} \rangle$ if and only if $\{i\}$ is a contrastive reason of $\langle f, \boldsymbol{x} \rangle$.*

*Proof.* For the first direction, let us begin by assuming that $\{i\}$ is a contrastive reason with respect to $\langle f, \mathbf{x} \rangle$. It then follows that $S \setminus \{i\}$ is not a sufficient reason for $\langle f, \mathbf{x} \rangle$. Consequently, any subset $S' \subseteq S \setminus \{i\}$ is also not a sufficient reason for $\langle f, \mathbf{x} \rangle$. In other words, this implies that for every subset $S \subseteq \{1, \ldots, n\}$ we have that $\text{suff}(f, \mathbf{x}, S \setminus \{i\}) = 0$. As a direct consequence, for any $S \subseteq \{1, \ldots, n\}$ the following condition holds: $\text{suff}(f, \mathbf{x}, S) = 1 \rightarrow \text{suff}(f, \mathbf{x}, S \setminus \{i\}) = 0$.

For the second direction, let us assume that $i$ is necessary with respect to $\langle f, \mathbf{x} \rangle$. We now assume, by way of contradiction, that $\{i\}$ is not a contrastive reason for $\langle f, \mathbf{x} \rangle$. Therefore, it follows that $\{1, \ldots, n\} \setminus \{i\}$ is a sufficient reason for $\langle f, \mathbf{x} \rangle$. Moreover, it clearly holds that the entire set $\{1, \ldots, n\}$ is a sufficient reason with respect to $\langle f, \mathbf{x} \rangle$ (fixing all features necessarily determines that the prediction remains the same). Overall, we get that:

$$\text{suff}(f, \mathbf{x}, \{1, \ldots, n\}) = 1 \wedge \text{suff}(f, \mathbf{x}, \{1, \ldots, n\} \setminus \{i\}) = 1 \tag{3}$$

This is in contradiction to the assumption that $i$ is necessary with respect to $\langle f, \mathbf{x} \rangle$.

**Theorem 2** *Any global sufficient reason of $f$ intersects with all local contrastive reasons of $\langle f, \boldsymbol{x} \rangle$ and any global contrastive reason of $f$ intersects with all local sufficient reasons of $\langle f, \boldsymbol{x} \rangle$.*

*Proof.* For the first part, let us assume, for the sake of contradiction, that there exists some global sufficient reason $S$ of $f$ and some local contrastive reason $S'$ of $\langle f, \mathbf{x} \rangle$ for which it holds that $S \cap S' = \emptyset$. Given that $S \cap S' = \emptyset$, it naturally follows that $S' \subseteq \bar{S}$. Taking into account that $S$ is a global

sufficient reason of $f$, we can infer that $S$ is also a local sufficient reason of $\langle f, \mathbf{x} \rangle$. Therefore, $\bar{S}$ does not qualify as a contrastive reason with respect to $\langle f, \mathbf{x} \rangle$, leading to the implication that no subset of $S$ can be a contrastive reason either. This assertion, however, contradicts the previously established $S' \subseteq \bar{S}$.

The second part of the claim will be almost identical to the first part: we can again assume, by way of contradiction, that there exists some global *contrastive* reason $S$ of $f$ and some local *sufficient* reason $S'$ of $\langle f, \mathbf{x} \rangle$ for which it holds that: $S \cap S' = \emptyset$. Given that $S \cap S' = \emptyset$, it naturally follows that $S' \subseteq \bar{S}$. Since $S$ is a global contrastive reason of $f$ it also acts as a local cntrastive reason for $\langle f, \mathbf{x} \rangle$. As a consequence, $\bar{S}$ can not be a sufficient reason for $\langle f, \mathbf{x} \rangle$. This implies that no subset of $S$ can serve as a sufficient reason for $\langle f, \mathbf{x} \rangle$, creating a contradiction with the premise that $S' \subseteq \bar{S}$.

**Theorem 3** *The MHS of all local contrastive reasons of $\langle f, \boldsymbol{x} \rangle$ is a cardinally minimal global sufficient reason of $f$, and the MHS of all local sufficient reasons of $\langle f, \boldsymbol{x} \rangle$ is a cardinally minimal global contrastive reason of $f$.*

Given some $f$, we denote $\mathbb{S}$ as the set of all local sufficient reasons of $\langle f, \mathbf{x} \rangle$ and denote $\mathbb{C}$ as the set of all local contrastive reasons of $\langle f, \mathbf{x} \rangle$. As a direct consequence of theorem 2, we can determine the following claim:

**Lemma 1** *A subset $S$ is a global sufficient reason of $f$ if and only if $S$ is a hitting set of $\mathbb{S}$ and is a global contrastive reason of $f$ if and only if $S$ is a hitting set of $\mathbb{C}$.*

As a consequence of Lemma 1, it directly follows that cardinally minimal local contrastive reasons are with correspondence to MHSs of $\mathbb{S}$, and cardinally minimal local sufficient reasons are with correspondence to MHSs of $\mathbb{C}$.

**The importance of the MHS duality.** An essential finding when dealing with inconsistent sets of clauses lies in a similar MHS duality between Minimal Unsatisfiable Sets (MUSes) and Minimal Correction Sets (MCSes) (Birnbaum & Lozinskii (2003); Bacchus & Katsirelos (2015)). In this context, MCSes are MHSs of MUSes, and vice versa (Bailey & Stuckey (2005); Liffiton & Sakallah (2008)). This discovery has played a pivotal role in the advancement of algorithms designed for MUSes and MCSes and this result has found applications in various contexts (Bacchus & Katsirelos (2015); Liffiton et al. (2016)). While the majority of this research focuses on propositional theories, others focus on Satisfiability Modulo Theories (SMT) (De Moura & Bjørner (2008)).

Within the context of explainable AI, previous research has shown similar duality relationships considering the relationship between *local* contrastive and sufficient reasons (Ignatiev et al. (2020b)) as well as the relationship between absolute sufficient reasons and adversarial attacks (Ignatiev et al. (2019)). This relationship was shown to be critical in the exact computation of local sufficient reasons for various ML models such as decision trees (Izza et al. (2022)), tree ensembles (Audemard et al. (2023)), and neural networks (Bassan & Katz (2023)).

## D   THE UNIQUENESS OF GLOBAL EXPLANATIONS

**Proposition 1** *There exists some $f$ and some $\boldsymbol{x} \in \mathbb{F}$ such that there are $2^{\lfloor \frac{n}{2} \rfloor}$ local subset minimal or cardinally minimal sufficient reasons of $\langle f, \boldsymbol{x} \rangle$.*

*Proof.* We construct $f$ as follows:

$$f(y) = \begin{cases} 1 & if \ \sum_{i=1}^{n} y_i \geq \lfloor \frac{n}{2} \rfloor \\ 0 & otherwise \end{cases} \tag{4}$$

We define the instance $\mathbf{x} := \mathbb{1}$. Clearly any subset $S$ of size $\lfloor \frac{n}{2} \rfloor$ or larger is a local sufficient reason of $\langle f, \mathbf{x} \rangle$ (since fixing the values of $S$ to $\mathbf{x}$ determines that the prediction remains: 1). Furthermore, every one of these subsets is *minimal* due to the fact that any subset of size $\lfloor \frac{n}{2} \rfloor - 1$ or smaller is *not* a sufficient reason of $\langle f, \mathbf{x} \rangle$ (it may cause a misclassification to class 0). Thus, it satisfies that there are $2^{\lfloor \frac{n}{2} \rfloor}$ subset minimal local sufficient reasons of $\langle f, \mathbf{x} \rangle$. Given that no local sufficient reason of size smaller than $\lfloor \frac{n}{2} \rfloor$ is present, these are, also *cardinally*-minimal sufficient reasons.

**Proposition 2** *If $S_1$ and $S_2$ are two global sufficient reasons of some non-trivial $f$, then $S_1 \cap S_2 = S \neq \emptyset$, and $S$ is a global sufficient reason of $f$.*

*Proof.* Our proof focuses on *non-trivial* functions, i.e., functions that do not always output $0$ or always output $1$. In other words, there exist some $\mathbf{x}, \mathbf{y} \in \mathbb{F}$ such that $f(\mathbf{x}) = 1$ and $f(\mathbf{y}) = 0$.

We begin by proving the following lemma:

**Lemma 2** *For any $f$ and $\boldsymbol{x} \in \mathbb{F}$, if $S$ is a sufficient reason of $\langle f, \boldsymbol{x} \rangle$ then there does not exist any $\mathbf{y} \in \mathbb{F}$ such that $f(\mathbf{y}) = \neg f(\boldsymbol{x})$ and there exists some $S' \subseteq \overline{S}$ that is a sufficient reason of $\langle f, \mathbf{y} \rangle$.*

*Proof.* Given that $S$ is sufficient for $\langle f, \mathbf{x} \rangle$, it follows that:

$$\forall (\mathbf{z} \in \mathbb{F}). \quad [f(\mathbf{x}_S; \mathbf{z}_{\bar{S}}) = f(\mathbf{x}) \neq f(\mathbf{y})] \tag{5}$$

By way of contradiction, let us assume that there exists some $\mathbf{y} \in \mathbb{F}$ for which there exists some $S' \subseteq \overline{S}$ that is a sufficient reason of $\langle f, \mathbf{y} \rangle$. This also implies that $\overline{S}$ is sufficient for $\langle f, \mathbf{y} \rangle$. In other words, the following condition holds:

$$\exists (\mathbf{y} \in \mathbb{F}), \; \forall (\mathbf{z} \in \mathbb{F}). \quad [f(\mathbf{y}_{\bar{S}}; \mathbf{z}_S) = f(\mathbf{y}) \neq f(\mathbf{x})] \tag{6}$$

Given that Equation 6 is valid for any $\mathbf{z} \in \mathbb{F}$, it is, consequently, applicable specifically to $\mathbf{x}$. In other words:

$$\exists (\mathbf{y} \in \mathbb{F}) \quad [f(\mathbf{y}_{\bar{S}}; \mathbf{x}_S) = f(\mathbf{y}) \neq f(\mathbf{x})] \tag{7}$$

This is inconsistent with the assertion that $S$ is sufficient for $\langle f, \mathbf{x} \rangle$.

**Lemma 3** *For a non-trivial function $f$, if $S$ is a sufficient reason of $\langle f, \boldsymbol{x} \rangle$ then any $S' \subseteq \overline{S}$ is not a global sufficient reason of $f$.*

*Proof.* Given that $S$ serves as a sufficient reason for $\langle f, \mathbf{x} \rangle$, it follows from Lemma 2 that there does not exist any $\mathbf{y} \in \mathbb{F}$ for which $f(\mathbf{y}) = \neg f(\mathbf{x})$ and $\overline{S}$ is not sufficient for $\langle f, \mathbf{y} \rangle$. Consequently, if there indeed exists some $\mathbf{y} \in \mathbb{F}$ for which $\overline{S}$ serves as a sufficient reason for $\langle f, \mathbf{y} \rangle$, it necessarily follows that $f(\mathbf{x}) = f(\mathbf{y})$.

Let us, for the sake of contradiction, assume the existance of some $S' \subseteq \overline{S}$ that serves as a global sufficient reason of $f$. This implication further entails that $\overline{S}$ is also a global sufficient reason for $f$. Consequently, $\overline{S}$ is also a local sufficient reason for $\langle f, \mathbf{y} \rangle$ for any $\mathbf{y} \in \mathbb{F}$. Given the property highlighted earlier, it holds that for any $\mathbf{y} \in \mathbb{F}$, we have $f(y) = f(\mathbf{x})$, which stands in contradiction to the premise that $f$ is non-trivial.

We are now in a position to prove the first part of proposition 2:

**Lemma 4** *If $S_1$ and $S_2$ are two global sufficient reasons of some non-trivial $f$, then $S_1 \cap S_2 \neq \emptyset$.*

*Proof.* Let us assume, to the contrary, that $S_1 \cap S_2 = \emptyset$. Hence, it follows that $S_1 \subseteq \overline{S_2}$. Given that $S_2$ is a global sufficient reason for $f$, it naturally follows that it is also a local sufficient reason for some $\langle f, \mathbf{x} \rangle$. However, Lemma 3 determines that there does not exist any $S' \subseteq \overline{S_2}$ that can be a global sufficient reason for $f$. This is in direct contradiction with our earlier inference that $S_1 \subseteq \overline{S_2}$ is a global sufficient reason for $f$.

We now can proceed to prove the second part of proposition 2:

**Lemma 5** *If $S_1$ and $S_2$ are global sufficient reasons of $f$, then $S = S_1 \cap S_2$ is a global sufficient reason of $f$.*

*Proof.* First, from Lemma 4, it holds that $S \neq \emptyset$. In instances where either $S_1 \subseteq S_2$ or $S_2 \subseteq S_1$, the claim is straightforwardly true. Therefore, our remaining task is to prove the claim for a non-empty set $S$ with the conditions $S \subsetneq S_1$ and $S \subsetneq S_2$.

Let us define the set $S' = \{1, \ldots, n\} \setminus \{S_1 \cup S_2\}$. Consider an arbitrary vector $\mathbf{x} \in \mathbb{F}$. Our aim is to prove that $S$ is a local sufficient reason with respect to $\langle f, \mathbf{x} \rangle$. Should this hold true for an arbitrary $\mathbf{x}$, it follows that $S$ constitutes a global sufficient reason of $f$.

Given that $S_1$ and $S_2$ are global sufficient reasons, it holds that:

$$\forall(\mathbf{z} \in \mathbb{F}). \quad [f(\mathbf{x}_{S_1}; \mathbf{z}_{\bar{S}_1}) = f(\mathbf{x}) = f(\mathbf{x}_{S_2}; \mathbf{z}_{\bar{S}_2})] \tag{8}$$

To demonstrate that $S$ is a local sufficient reason for $\langle f, \mathbf{x} \rangle$, let us assume, for the sake of contradiction, that it is not. Therefore, it satisfies that:

$$\begin{aligned} \exists(\mathbf{z} \in \mathbb{F}). \quad &[f(\mathbf{x}_S; \mathbf{z}_{\bar{S}}) \neq f(\mathbf{x})] \iff \\ \exists(\mathbf{z} \in \mathbb{F}). \quad &[f(\mathbf{x}_S; \mathbf{z}_{S_2 \setminus S}; \mathbf{z}_{\bar{S}_2}) \neq f(\mathbf{x})] \end{aligned} \tag{9}$$

Recall that $S_2$ is a global sufficient reason of $f$. Thus, assigning the features of $S$ to the corresponding values $\mathbf{x}$ and those of $S_2 \setminus S$ to $\mathbf{z}$ determines that the prediction remains the same (which in this case is *not* the value $f(\mathbf{x})$). Formally put:

$$\forall(\mathbf{z}' \in \mathbb{F}). \quad [f(\mathbf{x}_S; \mathbf{z}_{S_2 \setminus S}; \mathbf{z}'_{\bar{S}_2}) \neq f(\mathbf{x})] \tag{10}$$

This can be equivalently expressed as:

$$\forall(\mathbf{z}' \in \mathbb{F}). \quad [f(\mathbf{x}_S; \mathbf{z}_{S_2 \setminus S}; \mathbf{z}'_{S_1 \setminus S}; \mathbf{z}'_{S'}) \neq f(\mathbf{x})] \tag{11}$$

But we know that $S_1$ is a global sufficient reason and hence fixing the values of $S_1$ to $\mathbf{x}$ determines that the prediction is $f(\mathbf{x})$. Particularly, fixing the values of $S_1$ to $\mathbf{x}$ and the values of $S_2 \setminus S$ to $\mathbf{z}$ still determines that the prediction is always $f(\mathbf{x})$.

$$\forall(\mathbf{z}' \in \mathbb{F}). \quad [f(\mathbf{x}_{S_1}; \mathbf{z}_{S_2 \setminus S}; \mathbf{z}'_{S'}) = f(\mathbf{x})] \tag{12}$$

This particularly implies that:

$$\begin{aligned} \exists(\mathbf{z}' \in \mathbb{F}). \quad &[f(\mathbf{x}_{S_1}; \mathbf{z}_{S_2 \setminus S}; \mathbf{z}'_{S'}) = f(\mathbf{x})] \iff \\ \exists(\mathbf{z}' \in \mathbb{F}). \quad &[f(\mathbf{x}_S; \mathbf{z}_{S_2 \setminus S}; \mathbf{x}_{S_1 \setminus S}; \mathbf{z}'_{S'}) = f(\mathbf{x})] \end{aligned} \tag{13}$$

This contrasts with Equation 11.

**Theorem 4** *There exists one unique subset-minimal global sufficient reason of $f$.*

*Proof.* First, for the scenario in which $f$ is trivial (always outputs 1 or always outputs 0) it holds that any subset $S$ is a global sufficient reason. Therefore, $S = \emptyset$ is a unique subset-minimal global sufficient reason. Let us now consider a non-trivial function $f$. For the sake of contradiction, let us assume that two distinct subset minimal global sufficient reasons of $f$ exist: $S_1 \neq S_2$. Since $S_1$ and $S_2$ are subset minimal, it clearly holds that $S_1 \not\subseteq S_2$ and $S_2 \not\subseteq S_1$. Moreover, from proposition 2 it can be asserted that $S_1$ and $S_2$ are not disjoint, i.e., $S_1 \cap S_2 \neq \emptyset$. Now, we can use proposition 5, and conclude that $S = S_1 \cap S_2$ is also a global sufficient reason of $f$. This clearly contradicts the subset minimality of $S_1$ and $S_2$.

**Proposition 3** *For any possible ordering of features in line 4.2 of Algorithm 2, Algorithm 2 converges to the same global sufficient reason.*

Since Algorithm 2 converges to a subset-minimal global sufficient reason, and there is only one unique subset-minimal global sufficient reason (Theorem 4), then iterating over any ordering of features in line 4.2 of Algorithm 2 will converge to the same subset.

**Proposition 4** *Let $S$ be the unique subset minimal global sufficient reason of $f$. For all $i$, $i \in S$ if and only if $i$ is locally necessary for some $\langle f, \mathbf{x} \rangle$, and $i \in \overline{S}$ if and only if $i$ is globally redundant for $f$.*

We begin by proving the first part of the claim:

**Lemma 6** *$S$ is a unique subset-minimal global sufficient reason of $f$ if and only if for any $i \in S$ it holds that $i$ is locally necessary for some $\langle f, \boldsymbol{x} \rangle$.*

*Proof.* For the first direction, assume $i$ is necessary for some $\langle f, \mathbf{x} \rangle$. Then, from Theorem 1, it holds that $\{i\}$ is contrastive for some $\langle f, \mathbf{x} \rangle$. Furthermore, the first duality theorem (Theorem 2), implies that each local contrastive reason intersects with each global sufficient reason. Hence, we conclude that $i \in S$.

For the second direction, suppose that $S$ is a unique subset-minimal global sufficient reason of $f$. Let there be some $i \in S$. Since $S$ is a *unique* subset minimal global sufficient reason, then $\{1, \ldots, n\} \setminus \{i\}$ is necessarily *not* a global sufficient reason. If this was so, then there would exist some subset $S' \subseteq \{1, \ldots, n\} \setminus \{i\}$ that is a subset-minimal global sufficient reason, contradicting the uniqueness of $S$.

Since $\{1, \ldots n\} \setminus \{i\}$ is not a global sufficient reason, there exist some $\mathbf{x}', \mathbf{z}' \in \mathbb{F}$ such that:

$$f(\mathbf{x}'_{\{1,\ldots n\} \setminus i}; \mathbf{z}'_i) \neq f(\mathbf{x}') \tag{14}$$

Thus, $\{i\}$ serves as a contrastive reason for $\langle f, \mathbf{x}' \rangle$ and from Theorem 1 we can infer that $i$ is necessary with respect to $\langle f, \mathbf{x}' \rangle$.

For the second part of the claim, we prove the following Lemma:

**Lemma 7** *Let $S$ be the unique subset-minimal global sufficient reason of $f$. Then $i$ is globally redundant if and only if $i \in \overline{S}$.*

*Proof.* For the first direction, let us assume that $i$ is globally redundant and assume, for the sake of contradiction, that $i \in S$. Given that $i$ is globally redundant for $f$ then it holds that for any $\mathbf{x} \in \mathbb{F}$: $\text{suff}(f, \mathbf{x}, S) = 1 \rightarrow \text{suff}(f, \mathbf{x}, S \setminus \{i\}) = 1$. Hence, $S \setminus \{i\}$ is also a global sufficient reason of $S$, contradicting the subset-minimality of $S$.

For the second direction, assume that $i \in \overline{S}$. From Lemma 6, it holds that $i$ is not locally necessary for any $\langle f, \mathbf{x} \rangle$. In other words, there does not exist any $\mathbf{x} \in \mathbb{F}$ for which $\text{suff}(f, \mathbf{x}, S) = 1 \rightarrow \text{suff}(f, \mathbf{x}, S \setminus \{i\}) = 0$. This implies that for any $\mathbf{x} \in \mathbb{F}$ it satisfies that $\text{suff}(f, \mathbf{x}, S) = 1 \rightarrow \text{suff}(f, \mathbf{x}, S \setminus \{i\}) = 1$, i.e., that $i$ is globally redundant with respect to $f$.

**Proposition 5** *Any feature $i$ is either locally necessary for some $\langle f, \boldsymbol{x} \rangle$ or globally redundant for $f$.*

Building upon Proposition 4, we can discern that the unique subset minimal sufficient reason $S$ of $f$ categorizes all features into two distinct categories: those that are necessary for some $\langle f, \mathbf{x} \rangle$ and those that are globally redundant for $f$.

## E    PROOF OF PROPOSITION 6

**Proposition 6** *G-CSR is (i) coNP-Complete for MLPs, (ii) in PTIME for FBDDs and (iii) coNP-Complete for Perceptrons*

**Lemma 8** *G-CSR is coNP-Complete for MLPs.*

*Proof.* Membership is straightforward and is obtained since we can guess some $\mathbf{x}, \mathbf{z} \in \mathbb{F}$ and validate whether it satisfies that $f(\mathbf{x}_S; \mathbf{z}_{\bar{S}}) \neq f(\mathbf{x})$. If so, $\langle f, S \rangle \notin$ G-CSR.

Given our forthcoming proof that the *G-CSR* query for Perceptrons is coNP-Hard, it follows straightforwardly that the same is true for MLPs. Nevertheless, we show how hardness can also be proved particularly for MLPs via a reduction from the (local) *CSR* explainability query for MLPs.

Given the tuple $\langle f, \mathbf{x}, S \rangle$ we construct an MLP $f'$ which satisfies the following conditions:

$$f'(y) = \begin{cases} f(y) & if \ (\mathbf{x}_S = y_S) \\ 1 & if \ (\mathbf{x}_S \neq y_S) \end{cases} \tag{15}$$

If $\langle f, \mathbf{x}, S \rangle \in \textit{CSR}$, then it satisfies that:

$$\forall (\mathbf{z} \in \mathbb{F}). \quad [f(\mathbf{x}_S; \mathbf{z}_{\bar{S}}) = f(\mathbf{x})] \tag{16}$$

Given that $f'(y) = f(y)$ holds for any input for which $\mathbf{x}_S = y_S$, then it also satisfies that:

$$\forall (\mathbf{z} \in \mathbb{F}). \quad [f'(\mathbf{x}_S; \mathbf{z}_{\bar{S}}) = f'(\mathbf{x})] \iff$$
$$\forall (\mathbf{x}, \mathbf{z} \in \mathbb{F}). \quad (\mathbf{x}_S = \mathbf{z}_S) \rightarrow [f'(\mathbf{x}_S; \mathbf{z}_{\bar{S}}) = f'(\mathbf{x})] \tag{17}$$

If $\mathbf{x}_S \neq y_S$ then it consequently holds that $f'(y) = 1$. This implies that:

$$\forall (\mathbf{x}, \mathbf{z} \in \mathbb{F}). \quad (\mathbf{x}_S \neq \mathbf{z}_S) \rightarrow [f'(\mathbf{x}_S; \mathbf{z}_{\bar{S}}) = f'(\mathbf{x}) = 1] \tag{18}$$

Overall, we arrive at that:

$$\forall (\mathbf{x}, \mathbf{z} \in \mathbb{F}). \quad [f'(\mathbf{x}_S; \mathbf{z}_{\bar{S}}) = f'(\mathbf{x})] \tag{19}$$

implying that $\langle f', S \rangle \in \textit{G-CSR}$.

If $\langle f, \mathbf{x}, S \rangle \notin \textit{CSR}$, then it satisfies that:

$$\exists (\mathbf{z} \in \mathbb{F}). \quad [f(\mathbf{x}_S; \mathbf{z}_{\bar{S}}) \neq f(\mathbf{x})] \tag{20}$$

Given that $f'(y) = f(y)$, it follows that for any input satisfying $\mathbf{x}_S = y_S$ the following condition is also met:

$$\exists (\mathbf{z} \in \mathbb{F}). \quad [f'(\mathbf{x}_S; \mathbf{z}_{\bar{S}}) \neq f'(\mathbf{x})] \tag{21}$$

implying that:

$$\exists (\mathbf{x}, \mathbf{z} \in \mathbb{F}). \quad [f'(\mathbf{x}_S; \mathbf{z}_{\bar{S}}) \neq f'(\mathbf{x})] \tag{22}$$

Thus, it holds that $\langle f', S \rangle \notin \textit{G-CSR}$.

**Lemma 9** *G-CSR can be solved in polynomial time for FBDDs.*

*Proof.* Let $\langle f, S \rangle$ be an instance. We describe the following polynomial algorithm: We enumerate pairs of leaf nodes $(v, v')$ that correspond to the paths $(\alpha, \alpha')$. Let us denote by $\alpha_S$ the subset of nodes from $\alpha$ that correspond to the features of $S$. Given the pair $(\alpha, \alpha')$, the algorithm checks if $\alpha$ and $\alpha'$ "match" on all features from $S$ (more formally, there do not exist two nodes $v_\alpha \in \alpha_S$ and $v_{\alpha'} \in \alpha'_S$ with the same input feature $i$ and with different output edges). If we find two paths $\alpha$ and $\alpha'$ that match on all features in $S$, and that have different labels (one classified as True and the other: False) the algorithm returns "False" (i.e., $S$ is not a global sufficient reason of $f$). If we do not encounter any such pair $(v, v')$, the algorithm returns True.

**Lemma 10** *G-CSR is coNP-Complete for Perceptrons.*

*Proof.* Membership is straightforward since we can simply guess some $\mathbf{x}, \mathbf{z} \in \mathbb{F}$ and validate whether it satisfies that $f(\mathbf{x}_S; \mathbf{z}_{\bar{S}}) \neq f(\mathbf{x})$. If so, $\langle f, S \rangle \notin \textit{G-CSR}$.

We now will proceed to prove that *G-CSR* is also coNP-hard, We first briefly describe how the problem of (local) *CSR* can be solved in polynomial time for perceptrons, as proven by Barceló et al. (2020). This will give better intuition for the hardness reduction for the global setting. Given

some $\langle f, \mathbf{x}, S \rangle$, recall that a Perceptron $f$ is defined as $f = \langle \mathbf{w}, b \rangle$, where $\mathbf{w}$ is the weight vector and $b$ is the bias term. Therefore, it is possible to obtain the exact value of $\sum_{i \in S} \mathbf{x}_i \cdot w_i$.

Then, for the remaining features in $\bar{S}$, one can linearly determine the $y$ assignments corresponding to the *maximal* and *minimal* values of $\sum_{i \in \bar{S}} y_i \cdot w_i$. The maximal value is obtained by setting $y_i := 1$ whenever $\mathbf{w}_i \geq 0$ and $y_i := 0$ whenever $\mathbf{w}_i = 0$. The minimal value is obtained respectively (setting $y_i := 1$ whenever $\mathbf{w}_i < 0$ and $y_i := 0$ whenever $\mathbf{w}_i \geq 0$). We are now equipped with the capability to compute the entire spectrum of potential values that may be realized by assigning the values of $S$ to $\mathbf{x}$. It is hence straightforward that $S$ is a (local) sufficient reason for $\langle f, \mathbf{x} \rangle$ if and only if this entire range is always positive or always negative. This can be determined by checking whether both the minimal possible value and maximal possible value are both positive or both negative which is equivalent to checking whether the maximal possible value is negative or the minimal possible value is positive. Formally put:

$$
\sum_{i \in S} \mathbf{x}_i \cdot w_i + max\{\sum_{i \in \bar{S}} y_i \cdot w_i + b \mid y \in \mathbb{F}\} \leq 0 \ \vee
$$
$$
\sum_{i \in S} \mathbf{x}_i \cdot w_i + min\{\sum_{i \in \bar{S}} y_i \cdot w_i + b \mid y \in \mathbb{F}\} > 0
$$

(23)

This can clearly be determined in linear time using the computation method outlined above. Note that we require a strict inequation on the second term since we assumed w.l.o.g that a zero weighted term is classified as $0$ (the negative weighted class) and not $1$ (the positive weighted class).

Now, for the global setting, we notice that $max\{\sum_{i \in \bar{S}} y_i \cdot w_i + b \mid y \in \mathbb{F}\}$ and $min\{\sum_{i \in \bar{S}} y_i \cdot w_i + b \mid y \in \mathbb{F}\}$ can still be computed in the same manner as above. However, one must verify that equation 27 is satisfied for *every* possible value $\mathbf{x}$. This, in turn, carries implications for the associated complexity. We show, indeed, that *G-CSR* for perceptrons is coNP-hard.

We reduce *G-CSR* for Perceptrons from $\overline{SSP}$, known to be coNP-Complete. *SSP* (subset-sum-problem) is a classic NP-Complete problem which is defined as follows:

---

**SSP (Subset Sum Problem)**:
**Input**: $\langle (z_1, z_2, \ldots, z_n), T \rangle$, where $(z_1, z_2, \ldots, z_n)$ is a set of *positive integers* and $T$, is the target integer.
**Output**: *Yes*, if there exists a subset $S' \subseteq \{1, 2, \ldots, n\}$ such that $\sum_{i \in S'} z_i = T$, and *No* otherwise.

---

For the case of $\overline{SSP}$, the language decides whether there does not exist a subset of features $S' \subseteq (1, 2, \ldots, n)$ for which $\sum_{i \in S'} z_i = T$

We reduce *G-CSR* for Perceptrons from $\overline{SSP}$. Given some $\langle (z_1, z_2, \ldots, z_n), T \rangle$ we construct a Perceptron $f := \langle \mathbf{w}, b \rangle$ where it holds that $\mathbf{w} := (z_1, z_2, \ldots, z_n) \cdot (\mathbf{w}_{n+1})$ ($\mathbf{w}$ is of size $n + 1$), where $\mathbf{w}_{n+1} := \frac{1}{2}$, and $b := -(T + \frac{1}{4})$. The reduction computes $\langle f, S := \{1, \ldots, n\} \rangle$.

Clearly, it holds that:

$$
max\{\sum_{i \in \bar{S}} y_i \cdot \mathbf{w}_i \mid y \in \mathbb{F}\} = max\{\frac{1}{2}, 0\} = \frac{1}{2}
$$

(24)

and that:

$$
min\{\sum_{i \in \bar{S}} y_i \cdot \mathbf{w}_i \mid y \in \mathbb{F}\} = min\{\frac{1}{2}, 0\} = 0
$$

(25)

If $\langle (z_1, z_2, \ldots, z_n), T \rangle \in \overline{SSP}$, there does not exist a subset $S' \subseteq S = \{1, 2, \ldots, n\}$ for which $\sum_{i \in S'} z_i = T$, put differently — for any subset $S' \subseteq S = \{1, 2, \ldots, n\}$ it holds that $\sum_{i \in S'} z_i > T$ or $\sum_{i \in S} z_i < T$. But since the values in $(z_1, z_2, \ldots, z_n)$ are *positive integers* then it also holds that for any subset $S'$ the following condition is met:

$$[\sum_{i \in S'} z_i > T + \frac{1}{4}] \ \lor \ [\sum_{i \in S'} z_i < T - \frac{1}{4}] \iff$$

$$[\sum_{i \in S'} \mathbf{x}_i \cdot w_i > T + \frac{1}{4}] \ \lor \ [\sum_{i \in S'} \mathbf{x}_i \cdot w_i < T - \frac{1}{4}] \iff \tag{26}$$

$$[\sum_{i \in S} \mathbf{x}_i \cdot w_i > T + \frac{1}{4}] \ \lor \ [\sum_{i \in S} \mathbf{x}_i \cdot w_i < T - \frac{1}{4}]$$

Thus, equation 26 is maintained for any possible instance $\mathbf{x} \in \mathbb{F}$. This implies that for any $\mathbf{x} \in \mathbb{F}$, fixing the values of $S = \{1, \ldots, n\}$ always maintains either a positive or a negative value for $f$, thus implying that $S$ is a global sufficient reason of $f$ and that $\langle f, S \rangle \in$ *G-CSR*.

If $\langle (z_1, z_2, \ldots, z_n), T \rangle \notin \overline{SSP}$, then there exists a subset $S' \subseteq S = \{1, 2, \ldots, n\}$ for which $\sum_{s_i \in S'} z_i = T$, implying that:

$$[T - \frac{1}{4}] \leq \sum_{i \in S'} z_i \leq [T + \frac{1}{4}] \iff$$

$$[T - \frac{1}{4}] \leq \sum_{i \in S} \mathbf{x}_i \cdot w_i \leq [T + \frac{1}{4}] \tag{27}$$

Hence, $S$ is not a global sufficient reason for $f$ and $\langle f, S \rangle \notin$ *G-CSR*. This concludes the reduction.

Hardness results for Perceptrons, clearly indicate coNP-hardness for MLPs.

## F   PROOF OF PROPOSITION 7

**Proposition 7** *G-MSR is (i) coNP-Complete for MLPs, (ii) in PTIME for FBDDs and (iii) coNP-Complete for Perceptrons*

**Lemma 11** *G-MSR is coNP-Complete for Perceptrons.*

*Proof.* **Membership.** Membership is derived from the fact that one can guess some $\mathbf{x}^1, \ldots \mathbf{x}^n \in \mathbb{F}$ and $\mathbf{z}^1, \ldots, \mathbf{z}^n \in \mathbb{F}$. We then can validate for every feature $i \in (1, \ldots, n)$ whether: $f(\mathbf{x}^i_{\{1,\ldots,n\} \setminus \{i\}}; \mathbf{z}^i_{\{i\}}) \neq f(\mathbf{x}^i)$. This will imply that $\{i\}$ is contrastive with respect to $\langle f, \mathbf{x}^i \rangle$ and from Theorem 1, $i$ is necessary with respect to $\langle f, \mathbf{x}^i \rangle$. Now, from Proposition 4 it holds that $i$ is contained in the unique global subset minimal sufficient reason of $f$ if and only if $i$ is necessary with respect to some $\langle f, \mathbf{x} \rangle$. Therefore, it is possible to validate whether $\langle f, k \rangle \notin$ *G-MSR* using a certificate that checks whether the number of features that satisfy: $f(\mathbf{x}^i_{\{1,\ldots,n\} \setminus \{i\}}; \mathbf{z}^i_{\{i\}}) \neq f(\mathbf{x}^i)$ is larger than $k$.

**Hardness.** For hardness, we perform a similar reduction to the one performed for *G-CSR* for Perceptrons and reduce *G-MSR* for Perceptrons from $\overline{SSP}$. Given some $\langle (z_1, z_2, \ldots, z_n), T \rangle$, construct a Perceptron $f := \langle \mathbf{w}, b \rangle$ where we define $\mathbf{w} := (z_1, z_2, \ldots, z_n) \cdot (\mathbf{w}_{n+1})$ (i.e., $\mathbf{w}$ is of size $n + 1$), where $\mathbf{w}_{n+1} := \frac{1}{2}$, and $b := -(T + \frac{1}{4})$. The reduction computes $\langle f, k := n \rangle$.

Consider that $\langle (z_1, z_2, \ldots, z_n), T \rangle \in \overline{SSP}$. Drawing upon Lemma 10, we can infer that $S = \{1, 2, \ldots, n\}$ constitutes a global sufficient reason of $f$. To put it differently, a subset exists — trivially of of size $k = n$ in this instance — that serves as a global sufficient reason of $f$. Consequently, $\langle f, k \rangle \in$ *G-MSR* for Perceptrons.

Assume $\langle (z_1, z_2, \ldots, z_n), T \rangle \notin \overline{SSP}$. We need to prove that there does not exist any global sufficient reason of $f$ of size $k$ or less. Since any subset containing a sufficient reason is a sufficient reason, it is enough to show that there does not exist any global sufficient reason of *exactly* size $k$. From Lemma 10 we indeed already know that $(z_1, z_2, \ldots, z_n)$ is *not* a sufficient reason in this case. However, we still need to prove that there does not exist any *other* sufficient reason of size $k$.

Let $j \neq n + 1$ be some feature and let $S := \{1, 2, \ldots, n + 1\} \setminus \{j\}$ be some subset of features. We prove that $S$ is not a global sufficient reason for $f$. Since any $z_j$ in $(z_1, z_2, \ldots, z_n)$ is a positive integer, and since $\mathbf{w}_{n+1} = \frac{1}{2}$ is also positive, then it holds that:

$$
\begin{aligned}
max\{\sum_{i \in \bar{S}} y_i \cdot w_i \mid y \in \mathbb{F}\} = max\{z_j, 0\} = z_j \ \wedge \\
min\{\sum_{i \in \bar{S}} y_i \cdot w_i \mid y \in \mathbb{F}\} = min\{z_j, 0\} = 0
\end{aligned}
\tag{28}
$$

This implies that that $S$ is a global sufficient reason of $f$ iff for any $\mathbf{x} \in \mathbb{F}$ it holds that:

$$
[\sum_{i \in S} \mathbf{x}_i \cdot w_i > T] \ \vee \ [\sum_{i \in S} \mathbf{x}_i \cdot w_i \leq T - z_j]
\tag{29}
$$

Within Lemma 10 we have already determined that if $\langle (z_1, z_2, \ldots, z_n), T \rangle \notin \overline{SSP}$, then there must exist a value $\mathbf{x} \in \mathbb{F}$ such that:

$$
\begin{aligned}
\sum_{i \in \{1, \ldots, n\}} \mathbf{x}_i \cdot w_i = T \iff \\
\sum_{i \in \{1, \ldots, n\} \setminus \{j\}} \mathbf{x}_i \cdot w_i = T - z_j \iff \\
\sum_{i \in \{1, \ldots, n, n+1\} \setminus \{j\}} \mathbf{x}_i \cdot w_i = T - z_j + \frac{1}{2}
\end{aligned}
\tag{30}
$$

Now, since $T$ and $z_j$ are positive *integers*, then from equation 30 it holds that there exists some instance $\mathbf{x} \in \mathbb{F}$ such that:

$$
T - z_j < \sum_{i \in \{1, \ldots, n, n+1\} \setminus \{j\}} \mathbf{x}_i \cdot w_i < T
\tag{31}
$$

From equation 29, this implies that $\{1, \ldots, n, n + 1\} \setminus \{j\}$ is *not* a global sufficient reason of $f$. Since there does not exist any $j$ for which $\{1, \ldots, n, n + 1\} \setminus \{j\}$ is a global sufficient reason of $f$ and since we have already determined that $\{1, \ldots, n\}$ is not a global sufficient reason of $f$, we are left with that there does not exist any global sufficient reason of size $k$, concluding the reduction.

**Lemma 12** *G-MSR is in PTIME for FBDDs.*

*Proof.* Since *G-CSR* is in PTIME for FBDDs, we can use Proposition 3 which states that algorithm 2 always converges to the unique global cardinally minimal sufficient reason after a linear number of calls checking whether $\text{suff}(f, S \setminus \{i\}) = 1$. Each one of these calls can be performed in polynomial time (since *G-CSR* is polynomial for FBDDs), so hence using algorithm 2, we can obtain the unique global cardinally minimal sufficient reason of $f$ in polynomial time, and return True if it's size is smaller or equal to $k$, and False otherwise.

**Lemma 13** *G-MSR is coNP complete for MLPs.*

Both Hardness and Membership results trivially derive from those described for Perceptrons.

## G    PROOF OF PROPOSITION 8

**Proposition 8** *G-FR is (i) coNP-Complete for MLPs, (ii) in PTIME for FBDDs and (iii) coNP-Complete for Perceptrons*

**Lemma 14** *G-FR is coNP-Complete for Perceptrons.*

*Proof.* Membership is established from the fact that one can guess some $\mathbf{x}, \mathbf{z} \in \mathbb{F}$ and validate whether: $f(\mathbf{x}_{\{1,\ldots,n\}\setminus\{i\}}; \mathbf{z}_{\{i\}}) \neq f(\mathbf{x})$. From Theorem 1, this condition holds if and only if $i$ is necessary with respect to $\langle f, \mathbf{x} \rangle$. Furthermore, Proposition 5 establishes that this situation is equivalent to $i$ being *not* globally redundant with respect to $f$, thereby implying $\langle f, i \rangle \notin$ *G-FR*.

Before proving hardness, we will make use of the following Lemma which is simply a refined version of Proposition 4:

**Lemma 15** *$S$ is a global sufficient reason of $f$ iff for any $i \in \overline{S}$, $i$ is globally redundant.*

*Proof.* $S$ is a global sufficient reason of $f$ if and only if there exists some $S' \subseteq S$ which is a subset minimal global sufficient reason of $f$. From Proposition 4, it holds that any feature $i \in \bar{S}'$ is globally redunant, and since $\bar{S} \subseteq \bar{S}'$, it satisfies that any feature $i \in \bar{S}$ is globally redundant.

We are now in a position to employ Lemma 15, from which we can discern that $S$ qualifies as a global sufficient reason of $f$ if and only if every $i \in \bar{S}$ is globally redundant. Consequently, we can leverage the reduction that was utilized for establishing the coNP-Hardness of *G-CSR* for Perceptrons, as detailed in Lemma 10.

In other words, we can reduce *G-FR* for Perceptrons from $\overline{SSP}$. Given some $\langle (z_1, z_2, \ldots, z_n), T \rangle$ we can again construct a Perceptron $f := \langle \mathbf{w}, b \rangle$ where $\mathbf{w} := (z_1, z_2, \ldots, z_n) \cdot (\mathbf{w}_{n+1})$ ($\mathbf{w}$ is of size $n + 1$), $\mathbf{w}_{n+1} := \frac{1}{2}$, and $b := -(T + \frac{1}{4})$. The reduction computes $\langle f, i := n + 1 \rangle$.

It has been established in Lemma 10 that $S = \{1, 2, \ldots, n\}$ serves as a global sufficient reason of $f$ if and only if no subset $S' \subseteq S = \{1, 2, \ldots, n\}$ exists for which $\sum_{i \in S'} z_i = T$. Moreover, due to Lemma 15, the set $S = \{1, 2, \ldots, n\}$ is a global sufficient reason of $f$ if and only if any feature in $\bar{S}$ is globally redundant. However, $\bar{S}$ is in our case simply $\{n + 1\}$. This leads to the conclusion that feature $n + 1$ is globally redundant, thereby concluding the reduction.

**Lemma 16** *G-FR is coNP-Complete for MLPs.*

Both Hardness and Membership proofs for Perceptrons also trivially hold for MLPs.

**Lemma 17** *G-FR can be solved in polynomial time for FBDDs.*

*Proof.* Let $\langle f, i \rangle$ be an instance. We describe the following polynomial algorithm: We enumerate pairs of leaf nodes $(v, v')$ that correspond to the paths $(\alpha, \alpha')$. We denote by $\alpha_S$ the subset of nodes from $\alpha$ that correspond to the features of $S$. Given the pair $(\alpha, \alpha')$ we check if $\alpha$ and $\alpha'$ "match" on all features from $\{1, \ldots, n\} \setminus \{i\}$ (more formally, there do not exist two nodes $v_\alpha \in \alpha_{\{1,\ldots,n\}\setminus\{i\}}$ and $v_{\alpha'} \in \alpha'_{\{1,\ldots,n\}\setminus\{i\}}$ with the same input feature $j$ and with different output edges). If we find two paths $\alpha$ and $\alpha'$ that (i) match on all features in $\{1, \ldots, n\} \setminus \{i\}$, (ii) *do not* match on feature $i$ (i.e., have different output edges), and (iii) have different labels (one is classified as True and the other: False) the algorithm returns "False" (i.e, $i$ is *not* redundant with respect to $f$). If we do not encounter any such pair $(v, v')$, the algorithm returns "True".

## H  PROOF OF PROPOSITION 9

**Proposition 9** *G-FN is (i) coNP-Complete for MLPs, (ii) in PTIME for FBDDs and (iii) in PTIME for Perceptrons*

**Lemma 18** *G-FN is coNP-Complete for MLPs.*

*Proof.* To obtain membership, given a feature $i$ that we aim to verify as globally necessary with respect to $f$, we can guess an instance $\mathbf{x} \in \mathbb{F}$ and determine whether:

$$f(\mathbf{x}_{\{1,\ldots,n\}\setminus\{i\}}; \neg\mathbf{x}_{\{i\}}) = f(\mathbf{x}) \tag{32}$$

In other words, we wish to validate whether fixing all features in $\{1, \ldots, n\} \setminus \{i\}$ to their values in $\mathbf{x}$, and negating only the value of feature $i$ (to be $\neg\mathbf{x}_i$) changes the prediction of $f(\mathbf{x})$. Clearly, this

holds if and only if $\{i\}$ is *not* a contrastive reason for $\langle f, \mathbf{x} \rangle$ and from Theorem 1 this holds if and only if $i$ is *not* necessary with respect to $\langle f, \mathbf{x} \rangle$. Put differently, there exists some $\mathbf{x} \in \mathbb{F}$ for which $i$ is not necessary with respect to $\langle f, \mathbf{x} \rangle$. This implies that $\langle f, i \rangle \notin$ *G-FN*.

For Hardness, we will make use of the following Lemma whose proof appears in the work of Barceló et al. (2020).

**Lemma 19** *If we have a Boolean circuit $B$, we can create an MLP $f_B$ in polynomial time that represents an equivalent Boolean function with respect to $B$.*

We now prove hardness by reducing from *TAUT*, a well-known coNP-Complete problem which is defined as follows:

---

**TAUT (Tautology)**:
**Input**: A boolean formula $\psi$
**Output**: *Yes*, if $\psi$ is a tautology and *No* otherwise.

---

Given some $\langle \psi \rangle$ with variables: $x_1, \ldots x_n$ we can construct a new boolean formula:

$$\psi' := \psi \vee (x_{n+1} \wedge \overline{x_{n+1}}) \tag{33}$$

We then can use Lemma 19 to transform it to an MLP $f$ and construct $\langle f, i := n+1 \rangle$.

If $\langle \psi \rangle \in$ *TAUT* then it holds that:

$$f(\mathbf{x}_{\{1,\ldots,n\}}; \mathbf{1}_{n+1}) = 0 \quad \wedge \quad f(\mathbf{x}_{\{1,\ldots,n\}}; \mathbf{0}_{n+1}) = 1 \tag{34}$$

where $\mathbf{1}_{n+1}$ and $\mathbf{0}_{n+1}$ denote that feature $n+1$ is set to either 1 or 0.

Hence, for any value $\mathbf{x} \in \mathbb{F}$ we can find a corresponding instance $\mathbf{z} \in \mathbb{F}$ such that:

$$f(\mathbf{x}_{\{1,\ldots,n\}}; \mathbf{z}_{\{n+1\}}) \neq f(\mathbf{x}) \tag{35}$$

This implies that the subset $\{n+1\}$ is contrastive with respect to any $\langle f, \mathbf{x} \rangle$ and from theorem 1, $n+1$ is necessary with respect to any $\langle f, \mathbf{x} \rangle$. Thus, it satisfies that $n+1$ is globally necessary with respect to $f$ and consequently, $\langle f, i \rangle \in$ *G-FN*.

Let us now consider the scenario where $\langle \psi \rangle \notin$ *TAUT*. Under this assumption, it follows that there exists a False assignment for $\langle x_1, \ldots, x_n \rangle$, rendering $\psi'$ false irrespective of the assignment to $\mathbf{x}_{n+1}$. To put it differently, this scenario satisfies the following condition:

$$f(\mathbf{x}_{\{1,\ldots,n\}}; \mathbf{1}_{n+1}) = 0 \quad \wedge \quad f(\mathbf{x}_{\{1,\ldots,n\}}; \mathbf{0}_{n+1}) = 0 \tag{36}$$

Thus, we can take an arbitrary vector $\mathbf{x}$ and set some other arbitrary vector $\mathbf{z}$ to be equal to $\mathbf{x}$ on the first $n$ features and negated on feature $n+1$. Both of these vectors will be labeled to class 0 and it thereby satisfies that:

$$\exists \mathbf{z}, \mathbf{x} \in \mathbb{F} \quad f(\mathbf{x}_{\{1,\ldots,n\}\setminus\{i\}}; \mathbf{z}_{\{n+1\}}) = f(\mathbf{x}) \tag{37}$$

We can thus conclude that $\{n+1\}$ is *not* a contrastive reason of $\langle f, \mathbf{x} \rangle$ and from theorem 1, this implies that $n+1$ is not necessary with respect to $\langle f, \mathbf{x} \rangle$. Particularly, $n+1$ is not globally necessary, consequently implying that $\langle f, i \rangle \notin$ *G-FN*.

**Lemma 20** *G-FN can be solved in polynomial time for FBDDs.*

*Proof.* Let $\langle f, i \rangle$ be an instance. We describe the following polynomial algorithm: We enumerate pairs of leaf nodes $(v, v')$ that correspond to the paths $(\alpha, \alpha')$. We denote by $\alpha_S$ the subset of nodes from $\alpha$ that correspond to the features of $S$. Given the pair $(\alpha, \alpha')$ we check if $\alpha$ and $\alpha'$ "match" on all features from $\{1, \ldots, n\} \setminus \{i\}$ (more formally, there do not exist two nodes $v_\alpha \in \alpha_{\{1,\ldots,n\}\setminus\{i\}}$ and $v_{\alpha'} \in \alpha'_{\{1,\ldots,n\}\setminus\{i\}}$ with the same input feature $j$ and with different output edges). If we find

two paths $\alpha$ and $\alpha'$ that (i) match on all features in $\{1, \ldots, n\} \setminus \{i\}$, (ii) *do not* match on feature $i$ (i.e., have different output edges), and (iii) have *the same* label (both classified as True, or both classified as False) the algorithm returns "False" (i.e., $i$ is *not* globally necessary with respect to $f$). If we do not encounter any such pair $(v, v')$, the algorithm returns "True".

Clearly, if the algorithm encounters two paths $(\alpha, \alpha')$ that satisfy these three conditions, then it can be concluded that $\{i\}$ is *not* contrastive with respect to any assignment $\mathbf{x}$ associated with $\alpha$ and $\alpha'$. From Theorem 1, this implies that $i$ is not necessary with respect to the corresponding instances of $\langle f, \mathbf{x} \rangle$. However, if no such pair was encountered, then there does not exist an input $\mathbf{x} \in \mathbb{F}$ for which $\{i\}$ is not contrastive. It hence holds that $\{i\}$ is contrastive for any $\langle f, \mathbf{x} \rangle$ and Theorem 1 thereby implies that $i$ is necessary with respect to any $\langle f, \mathbf{x} \rangle$.

**Lemma 21** *G-FN can be solved in linear time for Perceptrons.*

*Proof.* Given some $\langle f, i \rangle$ such that $f := \langle \mathbf{w}, b \rangle$ is some Perceptron, we can perform a similar process to the one described under Lemma 10 and calculate: $max\{\sum_{j \in \{1, \ldots, n\} \setminus \{i\}} y_j \cdot w_j + b \mid y \in \mathbb{F}\}$ as well as: $min\{\sum_{j \in \{1, \ldots, n\} \setminus \{i\}} y_j \cdot w_j + b \mid y \in \mathbb{F}\}$ in polynomial time. We now simply need to check whether there exists any instance $\mathbf{x} \in \mathbb{F}$ for which:

$$
\begin{aligned}
\mathbf{x}_i \cdot w_i + max\{ \sum_{j \in \{1, \ldots, n\} \setminus \{i\}} y_j \cdot w_j + b \mid y \in \mathbb{F}\} \leq 0 \ \vee \\
\mathbf{x}_i \cdot w_i + min\{ \sum_{j \in \{1, \ldots, n\} \setminus \{i\}} y_j \cdot w_j + b \mid y \in \mathbb{F}\} > 0
\end{aligned}
\tag{38}
$$

This condition can obviously be validated in polynomial time since there are only two possible relevant scenarios ($\mathbf{x}_i = 1$ or $\mathbf{x}_i = 0$). If this condtion holds for one of the two possibilities then there exists an instance $\mathbf{x} \in \mathbb{F}$ for which $\{1, \ldots, n\} \setminus \{i\}$ is a local sufficient reason of $\langle f, \mathbf{x} \rangle$. It thereby holds that $\{i\}$ is *not* a contrastive reason of $\langle f, \mathbf{x} \rangle$. Hence, we can use Theorem 1, and conclude that $i$ is not necessary with respect to $\langle f, \mathbf{x} \rangle$, thus implying that $i$ is also not globally necessary. Should equation 38 not hold, it follows that for any $\mathbf{x} \in \mathbb{F}$ the set $\{1, \ldots, n\} \setminus \{i\}$ does *not* constitute a local sufficient reason of $\langle f, \mathbf{x} \rangle$. This conveys that $\{i\}$ is a local contrastive reason for *any* $\langle f, \mathbf{x} \rangle$. Theorem 1 further establishes that $i$ is necessary for any $\langle f, \mathbf{x} \rangle$, and hence $i$ is consequently globally necessary.

## I    PROOF OF PROPOSITION 10

**Proposition 10** *G-CC is (i) #P-Complete for MLPs, (ii) in PTIME for FBDDs and (iii) #P-Complete for Perceptrons.*

**Lemma 22** *G-CC is #P-Complete for Perceptrons.*

For simplification, we follow common conventions (Barceló et al. (2020)) and prove that the global counting procedure for: $C(S, f) = |\{\mathbf{x} \in \mathbb{F}, \mathbf{z} \in \{0, 1\}^{|\overline{S}|}, f(\mathbf{x}_S; \mathbf{z}_{\bar{S}}) = f(\mathbf{x})\}|$ is #P-Complete, rather than $c(S, f)$. Clearly, it holds that: $C(S, f) = c(S, f) \cdot 2^{|\overline{S}| + n}$ and hence $c(S, f)$ and $C(S, f)$ are interchangeable.

**Membership.** Membership is straightforward since the sum: $|\{\mathbf{x} \in \mathbb{F}, \mathbf{z} \in \{0, 1\}^{|\overline{S}|}, f(\mathbf{x}_S; \mathbf{z}_{\bar{S}}) = f(\mathbf{x})\}|$ is equivalent to the sum of certificates $(\mathbf{x}, \mathbf{z})$ satisfying:

$$
\exists \mathbf{x} \in \mathbb{F}, \exists \mathbf{z} \in \{0, 1\}^{|\overline{S}|}, f(\mathbf{x}_S; \mathbf{z}_{\bar{S}}) = f(\mathbf{x})
\tag{39}
$$

which is of course polynomially verifiable.

**Hardness.** We reduce from (local) *CC* of Perceptrons, which is #P-Complete. Given some $\langle f, S, \mathbf{x} \rangle$, where $f := \langle \mathbf{w}, b \rangle$ is a Perceptron, the reduction computes $f(\mathbf{x})$ and if $f(\mathbf{x}) = 1$ constructs $f' := \langle \mathbf{w}', b' \rangle$ such that $b' := b + \sum_{i \in S}(\mathbf{x}_i \cdot w_i)$, and $\mathbf{w}' := (\mathbf{w}_{\bar{S}}, \delta)$, with $\delta := (\sum_{i \in \overline{S}} |w_i|) - b'$. $\mathbf{w}_{\bar{S}}$ denotes a partial assignment where all features of the subset $\bar{S}$ are drawn from the vector $\mathbf{w}$ (the

vector $\mathbf{w}'$ is of size $|\bar{S}| + 1$). If $f(\mathbf{x}) = 0$, the reduction constructs $f' := \langle \mathbf{w}', b' \rangle$ with the same $b'$ but with $\mathbf{w}' := (\mathbf{w}_{\bar{S}}, \delta')$, such that $\delta' := -(\sum_{i \in \overline{S}} |w_i|) - b' - 1$.

For both reduction scenarios ($f(\mathbf{x}) = 1$ or $f(\mathbf{x}) = 0$) we will demonstrate that given the *global* completion count $C(\emptyset, f')$ we can determine the *local* completion count of $c(S, f, \mathbf{x})$ in polynomial time. We do this by proving the following Lemma:

**Lemma 23** *Given the polynomial construction of $f'$ it satisfies that:*

$$C(S, f, \boldsymbol{x}) = \sqrt{\frac{1}{2} \cdot C(\emptyset, f') - 2^{2|\bar{S}|}} \tag{40}$$

We denote $m$ and $t$ as the *number* of assignments $\mathbf{x}' \in \{0, 1\}^{|\bar{S}|+1}$, for which $f'$ predicts 0 or 1. Formally:

$$m := \left| \left\{ \mathbf{x}' \in \{0, 1\}^{|\bar{S}|+1} \,\middle|\, f'(\mathbf{x}') = 1 \right\} \right| \quad , \quad t := \left| \left\{ \mathbf{x}' \in \{0, 1\}^{|\bar{S}|+1} \,\middle|\, f'(\mathbf{x}') = 0 \right\} \right| \tag{41}$$

Clearly, it holds that:

$$m + t = 2^{|\bar{S}|+1} \tag{42}$$

It also satisfies that:

$$C(S := \emptyset, f') = |\{\mathbf{x}' \in \{0,1\}^{|\bar{S}|+1}, \mathbf{z} \in \{0,1\}^{|\bar{S}|+1}, f'(\mathbf{x}'_S; \mathbf{z}_{\bar{S}}) = f'(\mathbf{x}')\}| = m^2 + t^2 \tag{43}$$

As a result of equations 42 and 43, it satisfies that:

$$C(\emptyset, f') = m^2 + (2^{|\bar{S}|+1} - m)^2 \tag{44}$$

This implies that the aforementioned values of $m/t$ obey the following quadratic relation:

$$\begin{aligned} m/t &= \frac{-(-2^{|\bar{S}|+2}) \pm \sqrt{(-2^{|\bar{S}|+2})^2 - 4 \cdot 2 \cdot (2^{2|\bar{S}|+2} - c(\emptyset, f'))}}{2 \cdot 2} \\ &= \frac{2^{|\bar{S}|+2} \pm \sqrt{2^{2|\bar{S}|+4} - 8 \cdot (2^{2|\bar{S}|+2} - c(\emptyset, f'))}}{4} \\ &= 2^{|\bar{S}|} \pm \sqrt{2^{2|\bar{S}| - (2^{2|\bar{S}|+1} - \frac{1}{2} \cdot c(\emptyset, f'))}} \end{aligned}$$

Accordingly, $m/t$ must obey the following condition:

$$m/t = 2^{|\bar{S}|} \pm \sqrt{\frac{1}{2} \cdot c(\emptyset, f') - 2^{2|\bar{S}|}} \tag{45}$$

We will start by proving the first part of the Lemma. Specifically, to establish that when $f(\mathbf{x}) = 1$ the following condition is satisfied:

$$C(S, f, \mathbf{x}) = \sqrt{\frac{1}{2} \cdot C(\emptyset, f') - 2^{2|\bar{S}|}} \tag{46}$$

We will first prove that when $f(\mathbf{x}) = 1$, then there are at least $2^{|\bar{S}|}$ vectors $\mathbf{x}' \in \{0, 1\}^{|\bar{S}|+1}$ for which $f'(\mathbf{x}') = 1$.

First, we assume that $\mathbf{x}'_{|\bar{S}|+1} = 1$. For all $\mathbf{x}' \in \{0, 1\}^{|S|+1}$ such that $\mathbf{x}'_{|\bar{S}|+1} = 1$ it holds that:

$$
\begin{aligned}
w' \cdot \mathbf{x}' + b' &= \\
w'_{\bar{S}} \cdot \mathbf{x}'_{\bar{S}} + w'_{|\bar{S}|+1} \cdot \mathbf{x}'_{|\bar{S}|+1} + b' &= \\
w'_{\bar{S}} \cdot \mathbf{x}'_{\bar{S}} + \delta + b' &= \\
w'_{\bar{S}} \cdot \mathbf{x}'_{\bar{S}} + \left( \left( \sum_{i \in \overline{S}} |w_i| \right) - b' \right) + b' &= \\
w'_{\bar{S}} \cdot \mathbf{x}'_{\bar{S}} + \sum_{i \in \overline{S}} |w_i| &\geq 0
\end{aligned}
\tag{47}
$$

Given that there are precisely $2^{|\bar{S}|}$ assignments where $\mathbf{x}'_{|\bar{S}|+1} = 1$, it can be inferred that there are at least $2^{|\bar{S}|}$ assignments for which $f'(\mathbf{x}') = 1$. Hence, the following condition holds:

$$
m = 2^{|\bar{S}|} + \sqrt{\frac{1}{2} \cdot C(\emptyset, f') - 2^{2|\bar{S}|}}
\tag{48}
$$

Consequently, the exact number of assignments with $\mathbf{x}'_{|S|+1} = 0$ that satisfy that $f'(\mathbf{x}) = 1$ is exactly:

$$
\sqrt{\frac{1}{2} \cdot C(\emptyset, f') - 2^{2|\bar{S}|}}
\tag{49}
$$

Furthermore, it holds that:

$$
\begin{aligned}
f'(\mathbf{x}'_{\bar{S}}; \mathbf{0}_{|\bar{S}|+1}) = w'_{\bar{S}} \cdot \mathbf{x}'_{\bar{S}} + 0 + b' &= \\
w_{\bar{S}} \cdot \mathbf{x}'_{\bar{S}} + b + \sum_{i \in S} (\mathbf{x}_i \cdot w_i) &= \\
w_{\bar{S}} \cdot \mathbf{x}'_{\bar{S}} + w_S \cdot \mathbf{x}_S + b &= f(\mathbf{x}_S; \mathbf{x}'_{\bar{S}})
\end{aligned}
\tag{50}
$$

Thus, it follows that the count of assignments for which $\mathbf{x}'_{|S|+1} = 0$ that satisfy $f'(\mathbf{x}) = 1$ precisely equals the number of assignments for which $f(\mathbf{x}_S; \mathbf{x}'_{\bar{S}}) = 1$. This is, in fact, equivalent to the *local completion count*: $C(S, f, \mathbf{x})$. Put differently, this implies that:

$$
C(S, f, \mathbf{x}) = \sqrt{\frac{1}{2} \cdot C(\emptyset, f') - 2^{2|\bar{S}|}}
\tag{51}
$$

We now turn our attention to proving the second part of the Lemma. Specifically, we show that in the scenario where $f(\mathbf{x}) = 0$, the following condition is satisfied:

$$
C(S, f, \mathbf{x}) = \sqrt{\frac{1}{2} \cdot C(\emptyset, f') - 2^{2|\bar{S}|}}
\tag{52}
$$

We will similarly begin by proving that, given $f(\mathbf{x}) = 0$, there exist at least $2^{|\bar{S}|}$ vectors $\mathbf{x}' \in \{0,1\}^{|S|+1}$ for which $f'(\mathbf{x}') = 0$.

First, we assume that $\mathbf{x}'_{|\bar{S}|+1} = 1$. Now, for all $\mathbf{x}' \in \{0,1\}^{|S|+1}$ such that $\mathbf{x}'_{|\bar{S}|+1} = 1$ it holds that:

$$
\begin{aligned}
w' \cdot \mathbf{x}' + b' &= \\
w'_{\bar{S}} \cdot \mathbf{x}'_{\bar{S}} + w'_{|\overline{S}|+1} \cdot \mathbf{x}'_{|\overline{S}|+1} + b' &= \\
w'_{\bar{S}} \cdot \mathbf{x}'_{\bar{S}} + \delta' + b' &= \\
w'_{\bar{S}} \cdot \mathbf{x}'_{\bar{S}} + \left( -(\sum_{i \in \overline{S}} |w_i|) - b' - 1 \right) + b' &= \\
w'_{\bar{S}} \cdot \mathbf{x}'_{\bar{S}} - \sum_{i \in \overline{S}} |w_i| - 1 &< 0
\end{aligned}
\tag{53}
$$

Given that there are precisely $2^{|\bar{S}|}$ assignments where $\mathbf{x}'_{|\bar{S}|+1} = 1$, it follows that there exist at least $2^{|\bar{S}|}$ assignments for which $f'(\mathbf{x}') = 0$. Consequently, the following is satisfied:

$$
t = 2^{|\overline{S}|} + \sqrt{\frac{1}{2} \cdot C(\emptyset, f') - 2^{2|\overline{S}|}}
\tag{54}
$$

Therefore, the number of assignments, where $\mathbf{x}'_{|S|+1} = 1$, that satisfy the condition $f'(\mathbf{x}) = 0$ is as follows:

$$
\sqrt{\frac{1}{2} \cdot C(\emptyset, f') - 2^{2|\overline{S}|}}
\tag{55}
$$

Given the aforementioned reasons, we can deduce again that: $f'(\mathbf{x}'_{\bar{S}}; \mathbf{0}_{|\bar{S}|+1}) = f(\mathbf{x}_S; \mathbf{x}'_{\bar{S}})$. Consequently, the number of assignments where $\mathbf{x}'_{|S|+1} = 0$ and $f'(\mathbf{x}) = 0$ coincides with those where $f(\mathbf{x}_S; \mathbf{x}'_{\bar{S}}) = 0$. This corresponds to the *local* completion count: $C(S, f, \mathbf{x})$ in this context. In other words, it again holds that:

$$
C(S, f, \mathbf{x}) = \sqrt{\frac{1}{2} \cdot C(\emptyset, f') - 2^{2|\overline{S}|}}
\tag{56}
$$

which concludes the reduction.

**Lemma 24** *G-CC is #P-Complete for MLPs.*

Proofs of membership and Hardness for Perceptrons will also clearly hold for MLPs.

**Lemma 25** *G-CC is in PTIME for FBDDs.*

*Proof.* Similarly to the proof of the complexity of *G-CC* for Perceptrons (Lemma 22), we will assume the normalized count $C(S, f)$ which is interchangeable with $c(S, f)$. Each leaf node $v$ of $f$ corresponds to some path $\alpha$. We denote by $\alpha_S$ the subset of nodes from $\alpha$ that correspond to the features of $S$. We suggest the following polynomial algorithm: We enumerate pairs of leaf nodes $(v, v')$ which correspond to paths $(\alpha, \alpha')$. Given the pair $(v, v')$, we perform a counting procedure iff there do not exist two nodes $v_\alpha \in \alpha_S$ and $v_{\alpha'} \in \alpha'_S$ with the same input feature $i$ and with *different* output edges. Intuitively, this means that $\alpha$ and $\alpha'$ do not match on the subset $S$.

We define w.l.o.g that $v$ corresponds to the counting procedure over $\mathbf{x} \in \mathbb{F}$ and that $v'$ corresponds to the counting procedure over $\mathbf{z} \in \{0,1\}^{|\overline{S}|}$. Therefore, for each counting procedure, we add $2^{n-|\alpha|} \cdot 2^{|\overline{S}| - |\alpha'_{\overline{S}}|}$. Upon completing the iteration across all pairs $(v, v')$, we derive $C(S, f)$.

## J    PROOF OF PROPOSITION 11

In this section, we present detailed proofs of several results pertaining to *local* complexity queries, which have been referenced throughout the paper. First, we will briefly reference the results from

previous studies presented in Table 1. The findings related to the complexity of the local queries: *CSR*, *MSR* and *CC* for Perceptrons, FBDDs, and MLPs are drawn from the work of Barceló et al. (2020). Local complexity results for the *FR* and *FN* queries in the case of FBDDs as well as the local complexity class of the *FR* query for MLPs is provided in the works of Huang et al. (2023) and Huang et al. (2021). We now obtain the remaining complexity results that were mentioned in Table 1:

**Proposition 11** *(i) (Local) FN is in PTIME for Perceptrons and MLPs and (ii) (local) FR is coNP-Complete for Perceptrons.*

**Lemma 26** *FN is in PTIME for Perceptrons and MLPs*

Building upon the correctness of Theorem 1, we can deduce that determining the necessity of feature $i$ in relation to $\langle f, \mathbf{x} \rangle$ aligns with verifying if $\{i\}$ serves as a contrastive reason for $\langle f, \mathbf{x} \rangle$. For both MLPs and Perceptrons, it is possible to compute both $f(\mathbf{x}_{\{1,\ldots,n\}\setminus\{i\}}; \mathbf{1}_{\{i\}})$ and $f(\mathbf{x}_{\{1,\ldots,n\}\setminus\{i\}}; \mathbf{0}_{\{i\}})$ and validate whether:

$$f(\mathbf{x}_{\{1,\ldots,n\}\setminus\{i\}}; \mathbf{1}_{\{i\}}) \neq f(\mathbf{x}_{\{1,\ldots,n\}\setminus\{i\}}; \mathbf{0}_{\{i\}}) \tag{57}$$

The given condition is satisfied if, and only if, $\{i\}$ is contrastive with respect to $\langle f, \mathbf{x} \rangle$, thereby ascertaining whether $i$ is necessary in relation to $\langle f, \mathbf{x} \rangle$.

**Lemma 27** *FR is coNP-Complete for Perceptrons*

*Proof.* **Membership.** We recall that validating whether a subset $S$ is a local sufficient reason with respect to some $\langle f, \mathbf{x} \rangle$ can be done in polynomial time for Perceptrons, as was elaborated on in Lemma 10. This can be done by polynomially calculating both: $max\{\sum_{j\in\bar{S}} y_j \cdot w_j + b \mid y \in \mathbb{F}\}$ and $min\{\sum_{j\in\bar{S}} y_j \cdot w_j + b \mid y \in \mathbb{F}\}$ and then validating whether it holds that:

$$\mathbf{x}_i \cdot w_i + max\{\sum_{j\in\bar{S}} y_j \cdot w_j + b \mid y \in \mathbb{F}\} \leq 0 \ \lor$$
$$\mathbf{x}_i \cdot w_i + max\{\sum_{j\in\bar{S}} y_j \cdot w_j + b \mid y \in \mathbb{F}\} > 0 \tag{58}$$

Hence, membership in coNP holds since we can guess some subset $S \subseteq \{1, \ldots, n\}$ and polynomially validate whether it holds that:

$$\text{suff}(f, S, \mathbf{x}) = 1 \ \land \ \text{suff}(f, S \setminus \{i\}, \mathbf{x}) = 0 \tag{59}$$

If the following condition holds, then it satisfies that $i$ is not redundant with respect to $\langle f, \mathbf{x} \rangle$ and hence $\langle f, i \rangle \notin$ *FR*.

**Hardness.** We reduce *FR* for Perceptrons from the subset sum problem (SSP), specifically from $\overline{SSP}$ which is coNP-Complete. Given some $\langle (z_1, z_2, \ldots, z_n), T \rangle$ construct a Perceptron $f := \langle \mathbf{w}, b \rangle$ where we set $\mathbf{w} := (z_1, z_2, \ldots, z_n) \cdot (\mathbf{w}_{n+1})$ ($\mathbf{w}$ is of size $n + 1$), where $\mathbf{w}_{n+1} := \frac{1}{2}$, and $b := -(T + \frac{1}{4})$. The reduction computes $\langle f, \mathbf{x} := \mathbb{1}, i := n + 1 \rangle$.

Assume that $\langle (z_1, z_2, \ldots, z_n), T \rangle \in \overline{SSP}$. This implies that there does not exist any subset $S \subseteq \{1, \ldots, n\}$ for which $\sum_{j\in S} z_j = T$. Given that the values in $(z_1, \ldots, z_n)$ are *integers*, it consequently follows that there does not exist a subset $S$ satisfying that:

$$T - \frac{1}{2} < \sum_{j\in S} z_j < T + \frac{1}{2} \tag{60}$$

Consequently, it holds that there is no subset $S$ for which:

$$T - \frac{1}{2} < \sum_{j \in S} \mathbf{w}_j \cdot 1 < T + \frac{1}{2} \iff$$

$$-\frac{3}{4} < \sum_{j \in S} \mathbf{w}_j \cdot 1 + b < \frac{1}{4} \tag{61}$$

which is equivalent to:

$$[-\frac{3}{4} < \sum_{j \in S} \mathbf{w}_j \cdot 1 + \mathbf{w_{n+1}} \cdot 0 + b < \frac{1}{4}] \quad \wedge \quad [-\frac{1}{4} < \sum_{j \in S} \mathbf{w}_j \cdot 1 + \mathbf{w_{n+1}} \cdot 1 + b < \frac{3}{4}] \tag{62}$$

Therefore, no subset $S' \subseteq \{1, \ldots, n, n+1\}$ exists such that:

$$f(\mathbb{1}_{S'}; \mathbf{0}_{\bar{S}'}) \neq f(\mathbb{1}_{S' \setminus \{n+1\}}; \mathbf{0}_{\bar{S}' \cup \{n+1\}}) \tag{63}$$

Expressed differently, it can be asserted that:

$$\forall S' \subseteq \{1, \ldots, n, n+1\} \quad \text{suff}(f, S', \mathbb{1}) = 1 \rightarrow \text{suff}(f, S' \setminus \{n+1\}, \mathbb{1}) = 1 \tag{64}$$

Therefore, $n+1$ is redundant with respect to $\langle f, \mathbb{1} \rangle$, implying that $\langle f, \mathbb{1}, i \rangle \in$ *FR*.

Let us assume that $\langle (z_1, z_2, \ldots, z_n), T \rangle \notin \overline{SSP}$. From this assumption, it follows that there exists a subset of features, $S \subseteq \{z_1, \ldots z_n\}$ for which: $\sum_{j \in S} z_j = T$. We can express this equivalently as:

$$T = \sum_{j \in S} z_j \iff -\frac{1}{4} = \sum_{j \in S} \mathbf{w}_j + b \iff$$

$$[-\frac{1}{4} = \sum_{j \in S} \mathbf{w}_j + \mathbf{w}_{n+1} \cdot 0 + b] \quad \wedge \quad [\frac{1}{4} = \sum_{j \in S} \mathbf{w}_j + \mathbf{w}_{n+1} \cdot 1 + b] \tag{65}$$

We denote $S' := S \cup \{n+1\}$. Based on equation 65, we have that $f(\mathbb{1}_{S'}; \mathbf{0}_{\bar{S}'}) = 1$. Moreover, given that all features in $\bar{S}'$ are positive integers, it is also established that for any $S'' \subseteq \{1, \ldots, n+1\}$ for which $S' \subseteq S''$ the following holds: $f(\mathbb{1}_{S''}; \mathbf{0}_{\bar{S}''}) = 1$. Hence, $S'$ is sufficient with respect to $\langle f, \mathbb{1} \rangle$. Referring to equation 65, we observe that: $f(\mathbb{1}_{S' \setminus \{n+1\}}; \mathbf{0}_{\bar{S}' \cup \{n+1\}}) = 0$. This implies that $S' \setminus \{n+1\}$ is *not* sufficient with respect to $\langle f, \mathbb{1} \rangle$. In other words, we can conclude that:

$$\exists S' \subseteq \{1, \ldots, n, n+1\} \quad \text{suff}(f, S', \mathbb{1}) = 1 \wedge \text{suff}(f, S' \setminus \{n+1\}, \mathbb{1}) = 0 \tag{66}$$

Consequently, feature $n+1$ is *not* redundant with respect to $\langle f, \mathbb{1} \rangle$, thus implying that $\langle f, \mathbf{x}, i \rangle \notin$ *FR*.

