# OpenReview forum: "Local Vs. Global Interpretability: A Computational Perspective"
_ICLR.cc/2024/Conference — Submitted to ICLR 2024_

### Official Review · Reviewer_D69M · 2023-10-15

**Soundness:** 3 good
**Presentation:** 3 good
**Contribution:** 2 fair
**Rating:** 6
**Confidence:** 5

**Summary:**

The paper studies the computational complexity of different explainability queries on several classes of Boolean ML models. Since the picture for so-called "local" explaianability queries, that aim to explain the behavior of the model on a particular input, has been studied in depth in previous work by Barcelo et al, the paper focuses on "global" explainability queries, that aim to find the features that are most relevant for all inputs to the model. In particular, the paper aims to find whether there exists a correspondence between folklore claims on interpretability and the computational complexity of such problems.

**Strengths:**

- The paper is very well-written.
- It deals with a timely topic.
- Theoretical results unveil an interesting view on global explanations: sometimes they increase the complexity with respect to the corresponding local version of the explainability query, but sometimes they lower it.
- The observation that there is a unique minimal global sufficient reason is simple, but interesting.

**Weaknesses:**

- The paper lacks novelty, and as such I think it remains a notch below the acceptance threshold for ICLR. This is because the idea of correlating computational complexity and folklore claims on explainability of models has been proposed and studied before in a NeurIPS paper by Barcelo et al. So, while I liked the paper, I do not feel like championing it for a conference like ICLR.
- The paper lacks a unifying take-home message. In particular, it is never speculated or proposed a reason why for some models global explaianability is more difficult than local one, while for others is not. I could easily think of some potential reasons myself, so I am very surprised that the authors have not made it themselves.

**Questions:**

- Can you please propose a reason for why in some models, and for some queries, global explaianability is more complex than local one, while for others it's the opposite? This might be related to the characteristics of global explanations (an extra universal quantification in the definition of the notions vs a unique minimal global sufficient reason set) and on the nature of the models themselves (some problems being easily solvable on them).

- I would have been more positive about the paper if the authors would have considered a more relaxed version of global explainability, which seems to be of more practical interest: instead of requiring that the condition holds for every input instance, it could hold for a large fraction of them: say, for 90%, for instance. Can you add something about how your complexity analysis would change if this relaxed notion of global explainability was considered instead?

---

> ### Author Response · Authors · 2023-11-14
> **Response to Reviewer D69M**
>
> We thank the reviewer for the very valuable comments and constructive suggestions. Here is our response:
>
> [Weaknesses comments response]: The idea of leveraging computational complexity to evaluate interpretability has indeed been explored in earlier research, as well as in the work of Barcelo et al. As the reviewer points out, our study focuses on the traits and complexity classes of both local and global explainability queries. We believe that our research substantially extends the general understanding of this critical and timely subject, offering various novel insights and complexity results that shed light on the nature of local and global interpretability.
>
>
> We agree that certain enhancements to the manuscript are needed to better explain why some local queries are harder than global queries, or vice versa, and plan to incorporate these in the final version of the manuscript. Please see our detailed answers below.
>
> $\mathbf{Q1.}$ Can you please propose a reason for why in some models, and for some queries, global explainability is more complex than local one, while for others it's the opposite?
>
> $\mathbf{A1.}$ We agree that this is an important point, not sufficiently discussed in the paper - although we did touch on this briefly in section 5.2's proof sketches. In response to reviewer qXuS's feedback, we plan to condense the “proof sketches” section in our final version; and following this remark, we plan to use the space that will be freed to include a detailed analysis addressing this important point. Here is a short summary of the full and rigorous discussion we will include in the final version:
> 1. Linear models tend to have a lower complexity for local queries than global ones due to their structure. Each input feature $i$ in a linear model has an associated weight $w_i$. Let us assume that we seek to explain the *local* prediction of some input vector $x$. By multiplying a feature value $x_i$ with its corresponding weight $w_i$ we get $c_i = x_i \cdot w_i$, which can be seen as the exact direct contribution to the local prediction of feature $i$ to the prediction of $f(x)$. This ability to pinpoint each feature's contribution is the main core for constructing polynomial algorithms for solving *local* explainability queries. However, in the case of global queries, the difficulty in computation increases because the input $x$ is not included in the query's input; instead, only the weight $w$ is taken into account.
> 2. The disparity seen in linear models, where global queries are harder than local ones, does not apply to most queries in decision trees and neural networks. For decision trees, local explainability queries, which involve enumerating leaf nodes, can be extended to global queries by enumerating pairs of leaf nodes. In neural networks, the complexity found in encoding a boolean formula into an MLP is present in both local and global queries. However, in some queries, a surprising phenomenon happens — *local queries are actually harder than global ones*. A notable example is the minimum-sufficient-reason query. This phenomenon is apparent in this query due to the important property (proven in Section 4) of having a *unique* single global minimum sufficient reason, in contrast to a potentially *exponential* number of minimal local sufficient reasons, making the local task more complex than the global one. For similar reasons, this phenomenon also occurs in the redundant-feature query.
>
> We will provide a full and formal discussion of this question in the final version. We thank the reviewer for highlighting the importance of this point.
>
> $\mathbf{Q2.}$ I would have been more positive about the paper if the authors would have considered a more relaxed version of global explainability, which seems to be of more practical interest. Can you add something about how your complexity analysis would change if this relaxed notion of global explainability was considered instead?
>
> $\mathbf{A2.}$ The CC (Count-Completion) query discussed in our paper seeks to deal exactly with queries of this relaxed form. This query counts the exact portion of assignments that maintain the prediction, which translates to the probability of maintaining a prediction, assuming the input is independently and uniformly distributed. For example, if the global completion count is 0.9, this implies that 90% of the inputs maintain the prediction. In Proposition 10 of the appendix, we provide proofs of complexity classes for this explainability query. Interestingly, results show that for CC queries, the local and global forms of the query share the same complexity class. This is in stark contrast to the non-relaxed versions, where it is often observed that the local or global classes are strictly harder than the other. We agree with the reviewer's insight regarding the practicality of this query in certain scenarios compared to its non-relaxed counterpart. We intend to emphasize this analysis in our final version.

---

> > ### Comment · Reviewer_D69M · 2023-11-14
> >
> > Thanks for your clarifications. I appreciate the fact that you plan to restructure the paper and include the corresponding discussions about complexity and probabilistic explanations in the main body of the paper.  Based on this, I have increased my overall rating to a 6.

---

### Official Review · Reviewer_aaJG · 2023-10-30

**Soundness:** 3 good
**Presentation:** 4 excellent
**Contribution:** 3 good
**Rating:** 6
**Confidence:** 1

**Summary:**

This paper proposes a computational complexity theory perspective to evaluate the local and global interpretability of different ML models.
This framework examines various forms of local and global explanations and assesses the computational complexity involved in generating them.

**Strengths:**

As someone who primarily works in the field of explainability in machine learning, I have limited experience with computational complexity theory. However, I find the perspective of examining global and local interpretability through the lens of computational complexity both novel and interesting.

**Weaknesses:**

Due to my limited expertise in computational complexity theory, I have not delved deeply into the core aspects of the paper. My questions are high-level.

Could you clarify the claim that linear classifiers inherently possess local interpretability but lack global interpretability, whereas decision trees are acknowledged to have both local and global interpretability? In the XAI literature, a linear model is considered both locally and globally interpretable since it exhibits the same behavior everywhere. Could you clarify how this perspective on interpretability differs from yours?

Could you also highlight the differences between your work and the paper 'Model Interpretability through the Lens of Computational Complexity,' which seems to focus on local interpretability?

**Questions:**

Address questions in the previous section.

---

> ### Author Response · Authors · 2023-11-14
> **Response to Reviewer aaJG**
>
> We thank the reviewer for the comments!
>
> $\mathbf{Q1.}$ Could you clarify the claim that linear classifiers inherently possess local interpretability but lack global interpretability, whereas decision trees are acknowledged to have both local and global interpretability?
>
> $\mathbf{A1.}$ It is true that linear classifiers are typically considered to be generally interpretable (both locally and globally), in contrast to other more complex models such as neural networks.
> However, there are some subtle differences between local and global interpretability. In a general sense, the reason that linear classifiers are assumed to be interpretable is that the corresponding weight of each feature can be considered as a measure of the contribution of that specific feature. Assume that the weight vector of the linear classifier is some vector $w$ and that we are seeking a *local* explanation - i.e., the reason behind the prediction of a specific input vector $x$. We can calculate the exact contribution $c_i = x_i \cdot  w_i$, which signifies the contribution of feature $i$ to the local prediction over input $x$. However, if we consider *global* interpretability (how much does feature $i$ generally contribute to all possible input vectors $x$?) the task becomes more complex.
>
> This general observation was already pointed out in prior work [1]. However, in our work, we provide formal and rigorous indications that this is indeed the case, for some explanation contexts. For instance, while validating whether a subset of features is a local sufficient reason (CSR query) can be done in polynomial time for linear models, the same task becomes coNP-Complete when validating *global* sufficient reasons. A similar pattern occurs for other queries as well, such as feature redundancy. In other words, while validating whether a feature is locally redundant can be performed in polynomial time, validating whether a feature is globally redundant is coNP-Complete. We will try to make this point clearer in the text.
>
> [1] Interpretable Machine Learning - A Guide for Making Black Box Models Explainable; Molnar et al.
>
> $\mathbf{Q2.}$ Could you also highlight the differences between your work and the paper 'Model Interpretability through the Lens of Computational Complexity,' which seems to focus on local interpretability?
>
> $\mathbf{A2.}$ The idea of using computational complexity as a means to evaluate the interpretability of ML models is not new and was previously explored in a few different papers, including the one pointed out above. Our research, however, extends beyond the traditional scope of *local* explainability queries, by considering both local and global explainability queries. This broader focus allows us to offer fresh perspectives on the interpretability aspects of diverse ML models, such as linear models, decision trees, and neural networks, thereby enriching our understanding of both their local and global interpretability.
>
> Moreover, our research extends beyond the scope of the paper referenced above, by studying novel aspects of global explainability queries. We explore unique properties, such as the duality property and the uniqueness inherent in certain global explanation forms, which are integral to their corresponding complexity classes. Additionally, our study encompasses a broader range of queries. Beyond the fact that we study both local and global versions, we also investigate additional forms of explanations that include feature redundancy and feature necessity, which are related to notions of fairness.

---

### Official Review · Reviewer_RGYJ · 2023-10-31

**Soundness:** 3 good
**Presentation:** 3 good
**Contribution:** 3 good
**Rating:** 6
**Confidence:** 1

**Summary:**

The authors analyze the computational complexity of obtaining local and global explanations for different ML models. They study local and global explanations for linear models, decision trees and neural nets to evaluate the computational complexity associated with four different forms of explanations. They find that computing global explanations is computationally more difficult than computing local explanations for linear models. They also find that this reverses for decision trees and neural networks, i.e., computing a global explanation is more tractable than computing a local explanation for these models.

**Strengths:**

The work takes on the very challenging and impactful task of quantifying and measuring interpretability across models and types of explanations.

**Weaknesses:**

This is a theoretical analysis paper that seems to rely heavily on subsets of features that can be tractably enumerated. I do not think that this can practically extend to deep neural networks or to input domains like images and text.

**Questions:**

--

---

> ### Author Response · Authors · 2023-11-14
> **Response to Reviewer RGYJ**
>
> We thank the reviewer for the comments!
>
> Indeed, as the reviewer points out, the primary aim of our work is to deepen the theoretical and mathematical understanding of interpretability, while exploring the practical application of obtaining explanations for ML models is only a secondary aim.
>
> However, It is worth mentioning that there exists a line of recent work that focuses on a practical generation of explanations with formal guarantees by using formal reasoning techniques [1-6]. This can be done by leveraging different tools, such as Boolean Satisfiability (SAT) solvers, Satisfiability Modulo Theory (SMT) solvers, and Mixed-Integer Linear Programming (MILP) [1-4] solvers, as well as neural network verification techniques [5-6], whose scalability is improving rapidly [7].
>
> It is important to mention that this entire line of work focuses on the generation of *local* explanations with formal guarantees. We hence believe that our work, in addition to its theoretical value, lays a foundational framework for future empirical studies dedicated to crafting *global* explanations with formal guarantees, utilizing some of the aforementioned tools. We thank the reviewer for pointing this out and we will highlight these practical implications in the final version of our text.
>
> [1] Delivering Trustworthy AI through Formal XAI; AAAI 2022; Marques-Silva et al.
>
> [2] Abduction-based explanations for machine learning models; AAAI 2019; Ignatiev et al.
>
> [3] Explanations for Monotonic Classifiers; Neurips 2023; Marques-silva et al.
>
> [4] Explaining Naive Bayes and other linear classifiers with polynomial time and delay; Neurips 2020; Marques-Silva et al.
>
> [5] Verix: Towards Verified Explainability of Deep Neural Networks; Neurips 2023; Wu et al.
>
> [6] Towards Formal XAI: Formally Approximate Minimal Explanations of Neural Networks; TACAS 2023; Bassan et al.
>
> [7] Beta-crown: Efficient bound propagation with per-neuron split constraints for neural network robustness verification; Neurips 2021; Wang et al.

---

### Official Review · Reviewer_qXuS · 2023-11-01

**Soundness:** 3 good
**Presentation:** 2 fair
**Contribution:** 3 good
**Rating:** 6
**Confidence:** 3

**Summary:**

The paper presents a series of results about several forms of explanations (sufficient reasons, redundant and necessary features and completion count) both in terms of properties of these explanation forms and in terms of their computation complexity. The work focuses on boolean functions.
Interestingly, the paper delves into some existent duality between local and global explanations and provides uniqueness of minimal global sufficient reasons.
The authors further propose a notion of c-interpretability (where c may stand for computational?) which may be used to compare and assess the 'level of interpretability' of classes of models. The authors study decision trees, linear models and MLPs.

**Strengths:**

- The paper is very clear and notation and clear definitions are helpful to follow the work throughout
- I believe this work is highly significant, even if a bit narrow in the scope. Table 1 (and the results obtained to fill it) seem a valuable reference for researchers in the field.
- Overall, the content of the paper is of high value, and could serve to make some order (from a theoretical standpoint)  in the XAI field.
- Quality of result of Sec 3 and 4 is high (although please see my comments in Questions). I haven't carefully checked results of sec 5 as they are outside of my expertise.

**Weaknesses:**

- The work could be stronger if the connections to not-only boolean functions would be made clearer.
- *Paper & Appendix:* I found the proof sketches of limited use, if not misleading (e.g. that of proposition 2, checking the full proof in the appendix, I do not understand where's the link to Th. 2, as mentioned in the sketch). Proposition 1 could report the function from the appendix. I personally think space could be used better to comment on implications and reasons why the result is interesting/important.
Theorem 3 is not understandable given the lack of definition of hitting sets (in the paper); I'd suggest to either expand, or remove entirely. Global definitions of local counterparts could be suggested in a more straightforward way (as they all derive from local one by adding a
- Related work could be more comprehensive, especially when introducing the concepts of various explanations; see e.g. [1] and references therein.
- The introduced concept of the c-interpretability has unclear implications from a practical standpoint

References:
[1] Watson, David S., et al. "Local explanations via necessity and sufficiency: Unifying theory and practice." Uncertainty in Artificial Intelligence. PMLR, 2021.

**Questions:**

- Regarding Sec 4. Are these sentences true? If so, maybe consider add as commentary for additional clarity.
    1. Global subset-minimal and cardinally-minimal sufficient reasons coincide (thus you can talk about *the* minimal global sufficient reason, and can drop the "subset/carinally" identifiers)
  2. The minimal global sufficient reason can be characterized as the complementary of the union of all global redundant features.
- Isn't Proposition 5 a direct consequence of the definitions? If so, I'd suggest to put the comment inline and remove the proposition.

---

> ### Author Response · Authors · 2023-11-14
> **Response to Reviewer qXuS**
>
> We thank the reviewer for the valuable comments!
>
> $\mathbf{Q1.}$ Regarding Sec 4: Are these sentences true? If so, maybe consider add as commentary for additional clarity:
>
> $\mathbf{A1.}$ Indeed, both of these sentences are true, and stem from Theorem 4 and Proposition 4. We will clarify these statements in the text.
>
> $\mathbf{Q2.}$ Isn’t Proposition 5 a direct consequence of the definitions? If so, I’d suggest to put the comment inline and remove the proposition.
>
> $\mathbf{A2.}$ Proposition 5 is not a direct result of the definitions of local necessity and global redundancy. Hypothetically, there could perhaps be features that are neither locally necessary nor globally redundant, but we prove that such features cannot exist. We do agree though, that it is a direct corollary from the previous propositions (mainly Proposition 4). We will make this clearer in the text.
>
> [Weaknesses comments response]:
>
> $\mathbf{W1.}$ The work could be stronger if the connections to not-only boolean functions would be made clearer.
>
> $\mathbf{A1.}$ Following this suggestion, we will add a small section in the appendix that better emphasizes where exactly our results could be expanded to non-boolean functions, and where they could not. Basically, most theorems under sections 3+4 can be expanded to non-boolean cases as well as most of the hardness results under section 5.
>
> $\mathbf{W2.}$ Changes in Paper & Appendix:
>
> $\mathbf{A2.}$ Following this remark (and also reviewer D69M’s remark), we will significantly shorten the “proof-sketches” section 5.2 and replace it with a discussion on the topic of why some queries are computationally harder than others, within the context of global or local explanations.
> Moreover, we will fix the Theorem links you mentioned, introduce hitting sets in the main text, and put the definitions of global queries in the appendix.
>
> $\mathbf{W3.}$ Related work could be more comprehensive, especially when introducing the concepts of various explanations.
>
> $\mathbf{A3.}$ We will expand the `related work’ section and add additional references, including the ones mentioned in the review.
>
>
> $\mathbf{W4.}$ The introduced concept of the c-interpretability has unclear implications from a practical standpoint.
>
> $\mathbf{A4.}$ Indeed, the primary objective behind c-interpretability (computational interpretability) is theoretical and not practical. The idea is to employ the principles of computational complexity theory to methodically examine aspects of interpretability. In this context, interpretability is inversely related to complexity: the lower the complexity, the higher the interpretability, and vice versa.
>
> Nevertheless, it is worth mentioning that there is a line of work that focuses on a practical generation of explanations with formal guarantees. We elaborate on examples of such work in our response to reviewer RgYj. Since this previous line of work focused on *local* forms of explanations, we believe that our research, additionally to its theoretical significance, also establishes a basis for upcoming empirical research that aims to compute *global* explanations with formal guarantees. We appreciate the reviewer's insight on this matter and will emphasize these practical applications in our finalized manuscript.

---

### Author Response · Authors · 2023-11-14
**General Comments**

We thank all reviewers for their valuable comments and suggestions.

We are excited to see that reviewers appreciate this work and acknowledge its novel contributions in deepening our theoretical understanding of local and global interpretability in various ML models. We also thank the reviewers for their positive feedback regarding the writing style and overall presentation of our paper.

We will address each reviewer directly regarding questions, comments, and suggestions. However, we wanted to highlight here three main modifications that we intend to incorporate in the final draft, based on the reviewer’s insights:
1. In response to feedback from reviewers D69M and qXuS, we intend to shorten the proof sketches within section 5.2, and instead expand the discussion on the complexity classes of explainability queries, focusing on why local explanations are more complex than global ones, and the reverse for various queries.
2. Following the remark of reviewer D69M, we intend to add a more elaborate discussion regarding the *relaxed* global explainability form - the count-completion (CC) query, which may be more practical in some scenarios and is currently mainly discussed in the appendix.
3. Following the feedback from reviewers RGYj and qXuS, we will discuss the practical implications of our research, namely how this research can lay the groundwork for future studies focused on obtaining global explanations with formal guarantees, utilizing a range of formal reasoning methods.

---

### Public Comment · ~Shahaf_Bassan1 · 2024-06-06

We appreciate the further comments that were mentioned by the AC. Following the feedback provided by the AC and the reviewers, we have refined our paper, which can be accessed via https://arxiv.org/abs/2406.02981 and will be presented at the upcoming ICML conference.

---

### Meta-Review · Area_Chair_XUWL · 2023-12-14

**Metareview:**

This paper proposes to compare local vs. global explanations of classifiers against a proposed set of four key criteria: (1) sufficient reasons, (2) redundant features; (3) necessary features; (4) completion count.
The authors claim that this is the first work that performs computational analysis of global explanation methods.

Reviewer `qXuS` find the proof sketches to be of limited use and perhaps misleading. Reviewer `RGYJ` finds the work not applicable at all to deep neural networks as it requires the set of input features to be enumerable tractably and the authors agree with that.

Overall, the AC thinks the work has a **major flaw** because it has not referred to any existing global explanation methods, its actual forms and applications in interpretability, or any downstream tasks.
That should be the motivation for this work, which compares global against local explanations.

Therefore, the AC recommends `reject`.

**Justification For Why Not Higher Score:**

The work does not refer to any prior global explanation methods and therefore provides no context and motivation for comparing local and global explanations.

**Justification For Why Not Lower Score:**

N/A

---

### Decision · Program_Chairs · 2024-01-16

Reject